# Dredging and dumping impact coastal fluxes of sediment and organic carbon

**Lucas Porz** [1] ✉, **Jiayue Chen**[1], **Ruemeysa Yilmaz** [1], **Jannis Kuhlmann**[2], **Wenyan Zhang** [1] ✉ **& Corinna Schrum** [1,3]

The disturbance and relocation of coastal sediments by human activities constitutes a potentially large disruption to natural sediment and organic carbon fluxes, but large-scale estimates of these impacts are lacking. Here, we estimate the amounts of sediment and organic carbon disturbed and relocated in the North Sea resulting from (1) dredging in the form of mineral aggregate extraction and (2) material dumping during waterway maintenance. We show that despite disturbing less sediment than aggregate extraction, dumping causes greater carbon disturbance. We estimate carbon disturbance by both activities to be higher than by marine construction, but lower than by bottom-contacting fisheries. Simulations indicate that most dumped material re-deposits near the coast. Globally, dumping of organic carbon is estimated to 0.09–0.46 GtC yr$^{-1}$, and disturbance by material extraction to 0.04–0.08 GtC yr$^{-1}$. Comparison to natural processes suggests that these activities should be considered in regional to global sediment and carbon budgets.

As the interface between terrestrial and marine environments, coasts and marginal seas play an important role in regional to global sediment and carbon cycles. Therein, the seafloor acts both as a zone of transformation and long-term sequestration[1]. Recently, various budget estimates have aimed to quantify coastal particle and solute fluxes on regional or global scales by accounting for natural processes such as river loads, coastal erosion, redistribution by ocean currents, and sedimentation[2–4]. Studies of anthropogenic impacts on these fluxes have largely been limited to effects related to watershed modifications such as river damming, channel deepening, and erosion due to changes in climate and land-use[5–10], as well as associated impacts on ecosystem production and elemental cycles[11,12]. Several studies have also focused on the role of bottom-contacting fisheries in sediment and carbon disturbance[13–16]. In contrast, other activities that cause direct mechanical disturbance and/or relocation of sediment have received little attention.

When sediment is disrupted mechanically, organic matter within the suspension plumes can be transported, exposed to oxygenated waters and remineralised more readily than when trapped in the sediment, effectively increasing nutrient turnover rates and thereby reducing the efficacy of marine ecosystems as natural nutrient and carbon pumps ("Blue Carbon")[17]. In addition, seafloor disturbances can alter benthic habitats, with adverse impacts on ecosystem functions through changes to both macrobenthic and microbial communities[18–20].

In this study, we focus on two activities for which large-scale quantifications are lacking thus far: (1) aggregate extraction in the form of sand and gravel mining and (2) dumping of material originating from the dredging of waterways and harbours. We estimate the amounts of bulk sediment and organic carbon (OC) disturbed or relocated by dredging and dumping in the North Sea, a marginal sea under intense human pressures (Fig. 1). Based on publicly available data (see "Methods") and numerical simulations, we estimate amounts of disturbed bulk sediment and OC and potential OC remineralisation. We analyse the results with an emphasis on the Wadden Sea, Earth's largest tidal flat system that stretches along the Dutch, German, and Danish coasts. Dredging and dumping have been identified as main risks for maintaining the Outstanding Universal Value of the Wadden Sea World Heritage Site[21], with impacts including smothering of benthos, increased contaminant concentrations, and locally decreasing

[1]Institute of Coastal Systems, Helmholtz-Zentrum Hereon, Max-Planck-Strasse 1, Geesthacht, Germany. [2]BUND-Meeresschutzbüro, Bund für Umwelt und Naturschutz Deutschland e.V. (BUND), Bremen, Germany. [3]Institute of Oceanography, Center for Earth System Research and Sustainability, Universität Hamburg, Bundesstrasse 53, Hamburg, Germany. ✉e-mail: lucas.porz@hereon.de; wenyan.zhang@hereon.de

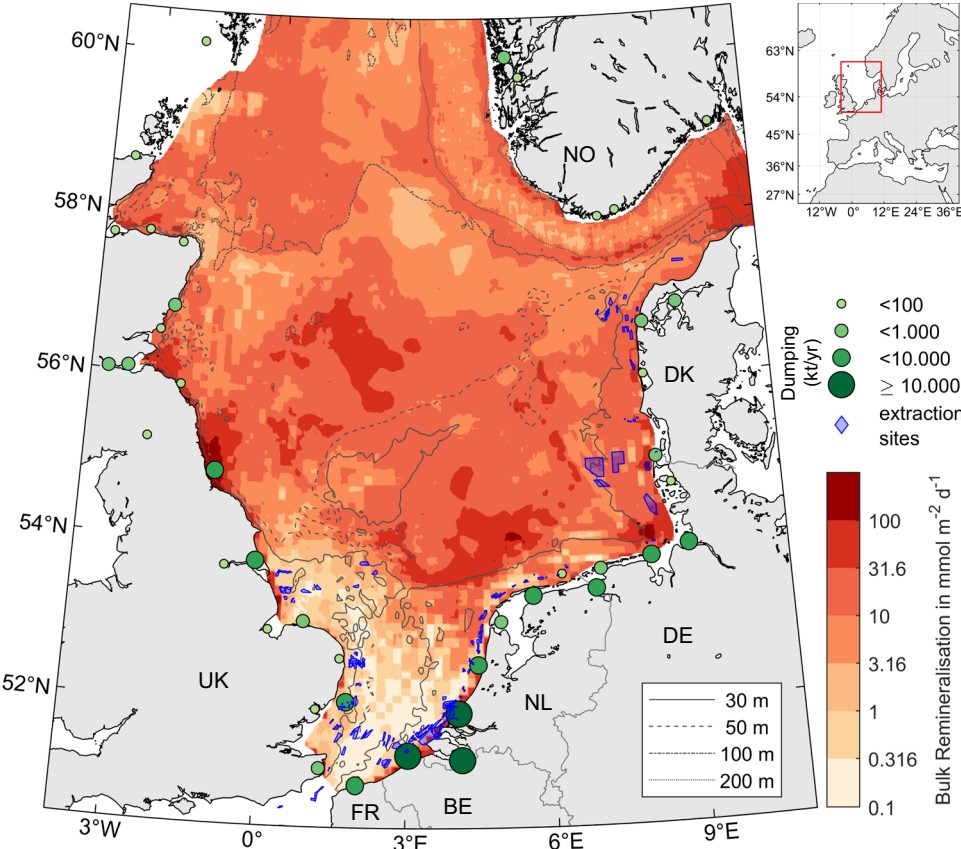

**Fig. 1 | Study area in the North Sea with dredging and dumping locations.** Annual bulk dumping activity averaged for 1995–2021 (green circles), and marine aggregate extraction areas (blue polygons). For visual representation, dumping amounts were cumulated on a 1° × 1° grid, and the dumping site coordinates for each grid cell averaged. Colour scale shows modelled sediment carbon mineralisation rates[24], with areas not covered by that model, including nearshore areas and those outside the study area, left blank. Projection: Albers Equal-Area Conic.

oxygen concentrations. By comparison to estimates of natural fluxes, we show that fluxes induced by humans can play a major role in shaping the overall sediment and carbon budgets of coastal oceans.

## Results

### Aggregate extraction

Our analysis (see "Methods") shows that in total, an order of 100,000 kt of marine aggregates in the form of sand and gravel are extracted each year in the North Sea for use in construction, land reclamation, and beach nourishment, with a slight increasing trend during the past three decades, albeit with considerable inter-annual variability (Fig. 2). In 2009 and 2010, the amount more than doubled compared to the multi-year average. The sudden increase in extraction starting in 2009 coincides with large coastal engineering efforts with high raw material requirements, such as the construction of the deep sea harbour *Jade-WeserPort* in Germany in 2009 and the Dutch sand engine, a large beach nourishment effort completed in 2011[22]. This indicates that individual projects can be responsible for the bulk of the disturbance through aggregate extraction.

The average concentration of sediment OC at the extraction sites is calculated to be 0.10% based on a published sediment carbon map[23]. We can therefore estimate the recent amount of annually disturbed OC from aggregate extraction in the North Sea to the order of 100 ktC. According to the TOCMAIM model[24], 87% of this disturbed OC is of low reactivity. The fractions of medium and high reactivity, which are more readily remineralised upon disturbance, make up 10.0% and 3.0% of OC, respectively. In our simulations, aggregate extraction results in an excess of about 1 ktC being remineralized after one year compared to the undisturbed scenario (see Supplementary Fig. 1), with minimal

difference between a scenario with continuous (1.12 ktC) and one with weekly (0.97 ktC) extraction, which corresponds to about 3.67 $ktCO_2$ released to the seawater (aqueous emissions) if all lost OC is remineralised to $CO_2$. The average relative reduction of sediment OC within extraction areas amounts to 5.7% within the upper 30 cm, being partially remineralised and partially resuspended and redistributed around the extraction sites (Supplementary Fig. 1).

It is important to note that the separation of OC into three reactivity classes is a simplification. It has been shown that OC of low reactivity may be made more bioavailable when exposed to oxygen, especially when mixed with more labile compounds ("priming")[25]. This priming effect is not well understood and was not included in our model, but has been estimated to result in a 23.4%–91.6% increase in bulk mineralisation rates[25]. Therefore, the longer-term potential of aggregate extraction to cause excess OC remineralisation may indeed be somewhat higher, but should not exceed aqueous emissions on the order of 10 $ktCO_2$ $yr^{-1}$. For comparison, the recent fuel emissions of the European dredging fleet was estimated at 600–800 $ktCO_2$ $yr^{-1}$ according to an industry report[26].

### Material dumping

Recent dumping of dredged material in the North Sea is 50,000–100,000 kt $yr^{-1}$, according to our analysis. Despite interannual variance, the data show no clear trend during the past decades, with the apparent onset of increase after 2008 attributable to the inclusion of data from the UK in that year. Our estimate for the amount of OC contained in dumped material is in the range of 500–5000 ktC $yr^{-1}$, reflecting the uncertainty in the OC content of dumped material (Fig. 3b).

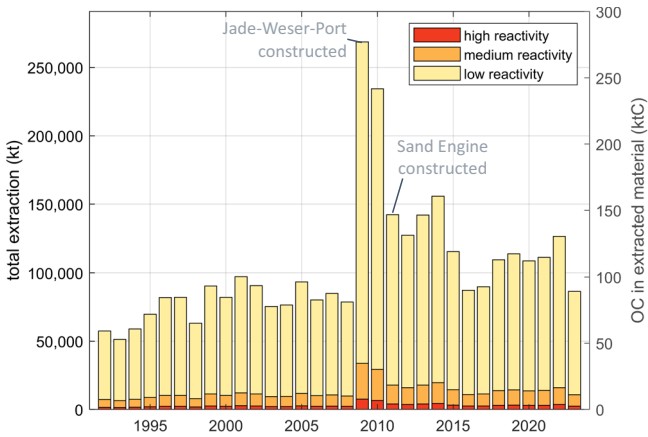

**Fig. 2 | Time series of aggregate extraction in the North Sea.** Total extracted amounts over time (left axis) with estimates of the disturbed sedimentary organic carbon of different reactivities according to the numerical model[24] (right axis).

In our dumping simulations, a large portion of the dumped fine-grained (and therefore OC-bearing) material remains close to the coast in both simulations with slowly sinking ($0.05\,mm\,s^{-1}$) and more rapidly sinking ($1\,mm\,s^{-1}$) particle tracers. Overall, 92.3–99.6% of dumped OC is deposited on the seabed after one year, while the remainder of is in suspension in the water column according to the simulation. Fig. 4 shows the resulting dumped OC concentration on the seabed after one year for assumed OC contents in dumped sediment of 5% in the fine fraction and 1% in the coarse fraction (see Supplementary Fig. 2 for full ranges). The slowly sinking tracers (Fig. 4a, b) distribute more toward the northeast along the direction of the residual circulation, while the rapidly sinking tracers (Fig. 4c) stay closer to the dumping locations. Simulated redistribution is also sensitive to whether dumped material is initially placed at the bottom or at the surface of the water column; when placed at the surface (Fig. 4a), transport farther toward the off-shore occurs, while the offshore distribution in the case of bottom-injected material (Fig. 4b) appears more gradual, with a smoother gradient and less patchiness.

The simulations reveal two distinct responses between the energetic Southern Bight and the more sheltered Wadden Sea; the Southern Bight is nearly completely winnowed by strong currents and associated high bottom-shear stresses in the shallow, coarse-grained connection to the Strait of Dover (see Supplementary Fig. 3), with some local deposition occurring near the muddy Belgian and Dutch coasts. The absence of stratification during much of the year, combined with intense tidal currents in the area, lead to recurrent deposition-resuspension cycles and a dispersion of the dumped material to considerable distances (>100 km) from the dumping sites. Meanwhile, dumped material does not spread as far from dumping sites in the Wadden Sea. A majority (59.4–69.2%) of the fine-grained sediment dumped within the German Bight and the Wadden Sea area deposits within the Wadden Sea World Heritage area, and 25.3–38.8% deposits within the 24 German Wadden Sea basins[27] (Table 1). Notably, 22.0–34.0% deposits at depths of <1.5 m, which corresponds to the mean low water level in the Wadden Sea and is roughly equivalent to its intertidal zone, meaning that dumping may contribute to the observed growth of intertidal flats[27] of the Wadden Sea.

As opposed to the Southern Bight, which is mainly non-depositional, the Wadden Sea is a net sink of sediment in the Southern North Sea, allowing a quantitative comparison between net import and redistribution by dumping. The amount of bulk sediment deposited in the Wadden Sea in the model after one year is on the order of 8000 kt, which is nearly the same amount as the total estimated mud deposition in the Wadden Sea ($7950\,kt\,yr^{-1}$) from riverine and marine sources[4]. This suggests that dumping can be as important for the internal distribution of mud in the Wadden Sea as natural sedimentation processes.

Estimating the potential remineralisation and corresponding climate impact of dumped material is complicated by technical and conceptual challenges. Firstly, a large proportion of the dumping sites are located close to or within the estuaries (Fig. 3a), where it is quickly transported back toward the harbours by tidal pumping as part of the residual estuarine circulation, and after which it may be dredged and dumped again. Though this phenomenon, which has been termed "circular dredging"[28], is a known issue in harbour management, it has not been well-documented in the scientific literature to date. Our simulations confirm this process, suggesting that a large part (roughly 25–40%) of dumped material is transported back to the nearshore and deposited there. This implies that the same material can be dredged and redeposited several times throughout the year and would thus be tallied repeatedly in the total quantities shown in Fig. 3b, even if the sediment budget of the overall system remained unchanged. This issue may also carry over to the dumping simulations, in which removal by dredging is not considered. As a result, the simulations may somewhat overestimate nearshore accumulation locally, but the dispersal and accumulation patterns outside of those dredged areas will not be affected. Therefore, the simulations give an indication of the overall transport patterns of dumped OC.

Secondly, the reactivity, i.e., the mineralisation rate of the OC in the dumped material, is largely unknown. Although it has been shown that large estuaries and harbour basins act as efficient bioreactors and thus OC exported from the estuary is largely inert and slowly microbially remineralised[29], dredged material may still be at the beginning of the mineralisation process and can therefore contain more reactive OC[30]. Nevertheless, our results allow an estimate of the lower and upper limits of remineralisation of 15 to 500 ktC, assuming that between 3% and 10% of the dumped OC is remineralised following dredging and dumping, which is the range of oxic remineralisation found in a 21-day incubation experiments using mud from the port of Hamburg[30]. If remineralised to $CO_2$, this would equate to emission from the seabed sediment to the water column of 55 to 1835 $ktCO_2\,yr^{-1}$.

## Comparison to other impacts

Table 2 lists the estimated impacts on sediment and OC in comparison to other sediment-disturbing activities, as well as natural sediment fluxes for which estimates have been compiled. Since the sandy seabed of the North Sea is mostly relict, and coastal erosion is the only considerable modern source of coarse-grained sediment to the North Sea, we compare mainly the fine-grained fluxes.

Both material extraction and dumping are at least equal in magnitude to the natural sediment fluxes. While large-scale OC fluxes in the North Sea are dominated by exchange with the North Atlantic, dumping potentially contributes as much OC as riverine export, and may be in the same magnitude as natural sedimentation. The largest total net impact on OC remineralisation is from bottom-contacting fisheries[15,24], estimated to aqueous emissions on the order of 1000 $ktCO_2\,yr^{-1}$. However, this impact has decreased during the last decade with an overall decrease of fishing effort in the area due to declining fish stocks and fishing quotas. The impact of aggregate extraction on net OC remineralisation is found to be comparatively low.

Most prominent among on-going marine infrastructure construction is the deployment of vast offshore windfarms in coastal waters globally. The initial disruption during the placement of pile foundations and cables, lasting days to weeks, is followed by an equilibration phase during which the ecosystem adjusts to the newly formed artificial reef over the next years to decades, and by a decommissioning phase, during which additional sediment disturbance may occur for days to weeks. For a typical twenty-year life cycle of commissioned and planned offshore wind farms in the southern North Sea, OC disturbance and remineralisation were

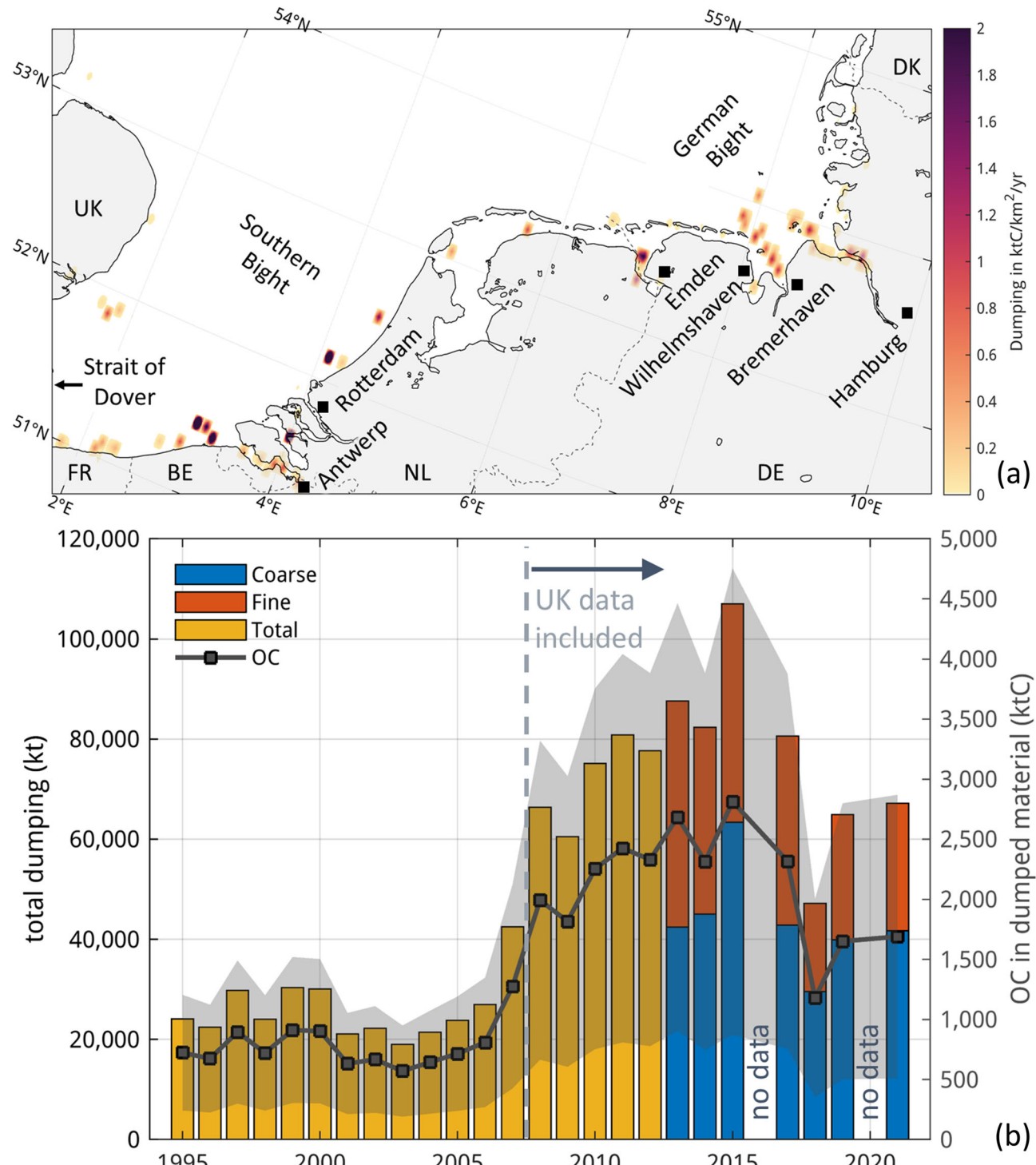

**Fig. 3 | Distribution and amount of material dumping in the North Sea.**
**a** Southeastern North Sea coast with estimated dumped organic carbon densities cumulated on a 3′ × 3′- grid and the locations of large ports (projection: oblique Mercator). **b** Dumped bulk amounts in the entire North Sea over time (left axis) and estimate of dumped organic carbon (right axis) with ranges indicated in grey shading based on OC content ranges in dumped material of 2–8% (midpoint: 5%) and 0–2% (midpoint: 1%) in the fine and coarse fractions, respectively.

estimated to 20 ktC yr$^{-1}$ and 20 ktCO$_2$ yr$^{-1}$, respectively[31], somewhat higher than our estimated impact for aggregate extraction and substantially less than for bottom-contacting fisheries and for disturbance by dumping. The same magnitude of OC disturbance as for wind farms has been estimated for the installation of subsea communication cables[32].

A comparison to human impacts on riverine fluxes is not straightforward, since contrasting effects on riverine export to the coast have been determined, with increased solute fluxes due to land use changes and decreased particulate fluxes due to retention by damming. In addition, riverine nutrient loads may play a more important role for the coastal carbon cycle than direct

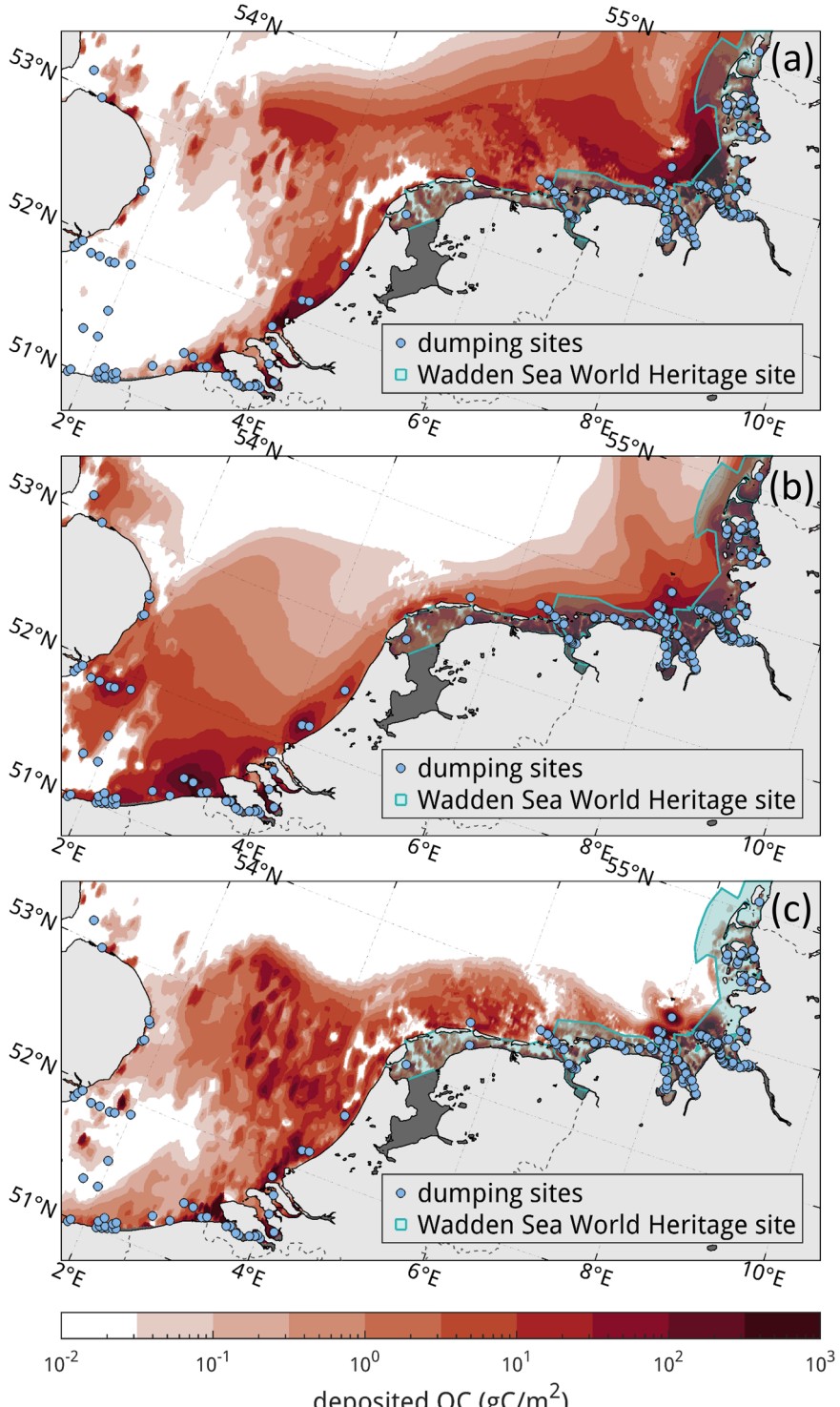

**Fig. 4 | Dumping simulation results.** Spatial distribution of dumped sediment organic carbon in the southeastern North Sea area after one year of continuous dumping with particle sinking velocities of (**a**, **b**) 0.05 mm s$^{-1}$ and (**c**) 1 mm s$^{-1}$ are shown. Dumped material is injected at the sea surface in (**a**, **c**) and at the seafloor in (**b**). Note the logarithmic colour mapping. Dumping locations are marked as blue circles and the Wadden Sea World Heritage site in blue shading. Areas in dark grey are not included in the model domain. Projection: oblique Mercator.

carbon exports by enhancing primary production and thereby increasing biological carbon fixation[12]. Considering the total OC export from rivers in the entire North Sea[33] of around 1100 ktC yr$^{-1}$ along with global estimates for the effect of damming (13% decrease[34]) and of European climate and land use change (20% increase[7]) brings the human impact on direct riverine export to the order of 100–200 ktC yr$^{-1}$, which is smaller than disturbances by both bottom-contacting

fisheries and dumping, and similar to the disturbance by aggregate extraction.

## Discussion
We show that dredging and dumping can impact the sediment fluxes of the North Sea at the same magnitude as natural processes of erosion and sedimentation. Based on these results, we argue that in heavily

**Table 1 | Dumped material distribution for different modelling scenarios**

|  | Scenario | | |
|---|---|---|---|
|  | Slowly sinking (0.05 mm/s), surface injection | Slowly sinking (0.05 mm/s), bottom injection | Rapidly sinking (1 mm/s), surface injection |
| Deposited, whole domain | 92.4 | 97.2 | 99.6 |
| Wadden Sea | 59.4 | 69.2 | 68.6 |
| Tidal basins | 25.3 | 38.1 | 38.8 |
| Intertidal area | 22.0 | 27.1 | 34.0 |

Values give the proportions of dumped material that has settled per area after one year of simulation for different modelling scenarios. The values for Wadden Sea, tidal basins[27] and intertidal area (depths <1.5 m) refer only to the sediment initially dumped within the Wadden Sea and German Bight. All values in percent.

**Table 2 | Sediment and organic carbon fluxes and disturbances in the North Sea in comparison**

|  |  | Sediment flux/ disturbance (kt yr$^{-1}$) | Organic carbon flux/ disturbance (ktC yr$^{-1}$) | Excess organic carbon remineralization (ktC yr$^{-1}$) | Recent trend | Projected trend | Sources |
|---|---|---|---|---|---|---|---|
| Human impact | Bottom-contacting fisheries | $10^6$ | $10^4$ | $10^2$ | ↓ | n.d. | 15,24 |
|  | Dumping | $10^4$ | $(0.5–5) \times 10^3$ | n.a. | → | ↑ | This study |
|  | Aggregate extraction | $10^5$ | $10^2$ | $(0.1–1) \times 10^1$ | → | ↑ | This study |
|  | Offshore wind farms* | n.d. | $10^1$ | $10^0$ | ↑ | ↑ | 31 |
|  | Telecommunication cables** | n.d. | $10^2$ | $10^0$ | ↑ | ↑ | 32 |
| Natural flux | Coastal erosion (incl. sand) | $(2–8) \times 10^3$ | n.d. | – |  |  | 71 |
|  | Aeolian deposition | $1.6 \times 10^3$ | n.a. |  |  |  | 72 |
|  | Riverine export | $(4.5–8.6) \times 10^3$ | $1.1 \times 10^3$ |  |  |  | 33,72,73 |
|  | Marine inflow | $(1.0–2.0) \times 10^4$ | $4.9 \times 10^4$ |  |  |  | 33,72,73 |
|  | Marine outflow | $(0.6–1.4) \times 10^4$ | $4.6 \times 10^4$ |  |  |  | 33,72,73 |
|  | Sedimentation | $(1.7–3.1) \times 10^4$ | $1.4 \times 10^3$ |  |  |  | 33,73,74 |

Estimated magnitudes of direct human impacts on bulk sediment disturbance, sediment organic carbon disturbance and potential sediment carbon remineralisation in the North Sea are listed with recent (decadal) and projected trends denoted as stagnant (→), increasing (↑), or decreasing (↓). Natural fluxes refer to suspended and dissolved loads only. n.d.: no data found; n.a.: not applicable.
*Southern North Sea only, assuming ~25% of disturbed carbon is remineralized[31].
**assuming 2 orders of magnitude lower impact than trawling[32].

managed systems, no sediment budget can be complete without consideration of the direct human impacts. This especially applies to depositional systems that tend to import sediment, such as coastal areas with growing tidal flats, lagoons, estuaries, and embayments. Such systems are likely to retain the signals of disturbed sediment and carbon for longer periods, whereas disturbances in sediment-exporting systems, often located at shallower, more open coasts, may be more quickly mixed and diluted by offshore transport with strong currents.

On a global scale, a considerable portion of the lateral sediment flux in coastal waters is due to human activities. The global mass of dredged and dumped material has been estimated to the order of 10 Gt annually[35], rivalling the export of riverine material to the coast (about 12.8 Gt yr$^{-1}$) in scale[36]. One global dataset[37] estimates that 4–8 Gt of sand and other sediments were dredged annually between 2012 and 2019 (Fig. 5). According to that data, countries with highest rates of dredging occurring within their Exclusive Economic Zones (EEZs) are the United States, China, the Netherlands, the United Kingdom, and Germany (Fig. 5a). Accounting for the size of each EEZ highlights regions with comparatively smaller EEZs and a higher concentration of dredging activity, such as the North and Baltic Seas, the Persian Gulf, and the Gulf of Guinea (Fig. 5b).

Assuming global dumping rates of 10 Gt yr$^{-1}$ and a similar composition of dumped material as in our study area with a range in OC content of 0.9–4.6% brings the mass of OC dumped at global coasts to 0.09 to 0.46 GtC yr$^{-1}$. This corresponds to, at minimum, 21% of the global OC flux by rivers to the coast[38], which has been estimated at 0.42 GtC yr$^{-1}$. The impact of material extraction on carbon fluxes is found to

be comparatively smaller. Assuming a similar composition as in our study area brings the global carbon disturbance from sand mining activities to 0.04–0.08 GtC yr$^{-1}$ and an excess mineralisation to 4–8 × 10$^{-4}$ GtC yr$^{-1}$.

Currently, the most substantial sediment carbon disturbances in the North Sea are assessed, in descending order of magnitude, as follows: bottom-contacting fisheries, material dumping, aggregate extraction, and marine construction. However, this sequence is subject to change in the coming decades, as certain activities, such as wind farm construction, increase at a rapid pace, while others, such as fishing, have shown decreasing trends. In addition, sea level rise is expected to enhance the sediment deficit in coastal regions such as the Wadden Sea, which will require additional dredging and sand nourishment to prevent the drowning of coastal landscapes[39]. Likewise, dumping of harbour mud is projected to increase with increasing vessel sizes, requiring deeper and wider navigation channels[40]. Other activities for which large-scale quantifications of corresponding sediment and carbon disturbances are yet to be quantified include anchor dragging, mooring and ore mining[41,42].

Potential climate impacts associated with these activities are less clear-cut. In contrast to aggregate extraction, dumping lacks the natural baseline to which it can be compared, because harbours and channels are artificial structures that require regular dredging to maintain their functioning. We therefore choose not to define remineralisation of dumped material as excess remineralisation. It is also important to note that the numbers for excess carbon remineralisation presented here should not be equated to atmospheric $CO_2$ emissions, as only a portion of remineralised OC will directly outgas due to the buffering capacity of

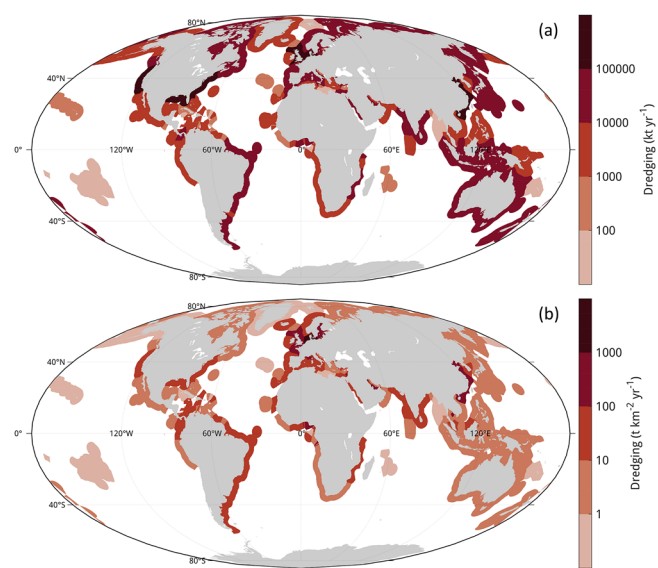

**Fig. 5 | Global dredging distribution.** Annually dredged amounts are averaged for 2012–2019 per Exclusive Economic Zone (EEZ). Maps show (**a**) total dredged amount, and (**b**) dredged amount per EEZ-area. Note the logarithmic colour mapping. Values extracted from the Marine Sand Watch platform (https://unepgrid.ch/en/marinesandwatch; last accessed 28/10/2025). Projection: Mollweide.

the seawater carbonate system[43]. Nevertheless, a corresponding effect on air-sea $CO_2$ exchange is likely, as eventual equilibration with the atmosphere is expected in the shallow, mixed coastal zone.

While this study gives first estimates of the human impacts on coastal sediment and carbon fluxes, further studies gauging impacts on the overall carbon cycle should include resolving indirect effects, such as changes to ecosystem production through resuspension of nutrients or habitat degradation. The impact of dumping on benthic habitats seems to be strongly area-specific, with some benthic communities showing high resilience, rapid recovery, and only local effects[44,45], while other areas show more long-lasting and wider spread impacts[46]. Our dumping simulations consider fine-grained sediments, but dumped material may also contain a substantial amount of coarse-grained sediment such as sand and gravel, which, though it contains little OC, may alter carbon fluxes indirectly through habitat alteration. Another underexplored mechanism regards alkalinity consumption through pyrite oxidation following sediment disturbance, which can further weaken the ocean's role as a carbon sink[47,48].

Thus far, management of sediment disturbance has not been considered in discussions around Blue Carbon potentials and crediting schemes, which have mainly centred around the restoration and protection of vegetated coastal ecosystems[49,50]. For example, the combined sequestration potential of marshes, seagrass and macroalgae in the German North Sea[49] has been estimated at 24–41 ktC yr⁻¹. Similar magnitudes are expected for the Blue Carbon potential of the other nations bordering the North Sea[51], all of them much lower than the amount of OC dumped and in the same magnitude as excess OC remineralisation from aggregate extraction. It has become clear that these processes cannot be ignored from a budgeting perspective, and future work should determine how they can be integrated within Blue Carbon policy frameworks.

In conclusion, it is evident that any holistic analysis of contemporary matter fluxes in the Earth system must account for direct human sediment disturbance and relocation.

## Methods
### Aggregate extraction data
The amount of material extracted from the European seabed has increased drastically, by two orders of magnitude since the 1970's[52].

Aggregate extraction is typically conducted using suction dredgers that pump a sediment-water mixture from the seafloor into hoppers onboard the vessels. The suction heads themselves create resuspension plumes at the seafloor. On the vessels, the excess, sediment-laden water spills back into the ocean as overflow discharge along with any unwanted, fine particulates that are filtered out, creating highly concentrated surface sediment plumes that can be detected up to several kilometres from the extraction sites[53]. These plumes of fine-grained sediment contain most of the organic matter present in the dredged material[54]. Extraction sites are concentrated in the southern North Sea and at the Danish coast (Fig. 1).

Data on 217 existing extraction areas[55] and extraction mass and/or volume[52,55,56] from 1992–2023 were analysed. Only extraction areas with a status classified as "Active" were considered, and a large shell extraction area spanning a large stretch of the Dutch coast was excluded from analysis.

In cases where extraction amounts were given in volume, they were converted to mass using the averaged porosity of all extraction sites of 0.36 and a grain density of 2650 kg m⁻³. Porosity was derived by interpolation of more than 2000 sediment samples collected in the region[23]. The amount of OC impacted was calculated for three OC classes of different reactivities based on results of a numerical model (see "Numerical models"). As an exact colocation of extracted amounts to extraction sites was not possible for all data points, we assumed that the extracted amount was evenly distributed across all extraction sites. For the same reason, it was not always possible to distinguish between extracted amounts inside and outside of the North Sea for the UK, Denmark, Germany, France, and Norway. Therefore, the total amounts may be overestimated since they also contain extraction from the English Channel and the Baltic, Irish and Celtic Seas. However, since the largest extraction sites are located within the North Sea, and the largest amounts by far are extracted from the Dutch coast, we do not expect this to affect the magnitude of our estimate.

### Dumping data
The majority of dumped material in the North Sea originates from maintenance dredging in navigation channels and harbour basins, with only a small amount of dredged material being disposed of on land[4]. Most dumping is concentrated at the southern North Sea coast near large ports such as those of Antwerp, Rotterdam, and Hamburg (Fig. 1).

Dumping data[57] compiled by the OSPAR Commission were analysed for information on dumping locations and quantities, containing data for 1995–2021, except for the years 2016 and 2020, as well as information on material composition (such as grain size) starting 2013. Beach nourishment has been included in monitoring efforts starting from 2013. No data for the UK is contained before 2008, and activities in Denmark and Norway appear from 2013.

Deposited amounts were located using deposit site codes for the years 2014, 2015, 2017–2019, and 2021 and filtered for the North Sea area. For the latter three years, no deposit site codes are included for activities in Denmark, and for 2018, no deposit site codes are included for some activities in Belgium, preventing the location of those activities. Though grain size was not specified for those activities, nearly all of them were classified as land reclamation, beach nourishment or construction, indicating that this dumped material comprised coarse-grained sediment. In total, 11,200, 3200 and 1600 kt of dumping are omitted in the years 2018, 2019 and 2021, respectively, amounting to a few percent of the multi-year average.

Because of the heterogenous nature of dumping data documentation, dumped material was categorised into three types based on sediment descriptions in the data: "coarse" for sand and coarser material (22.0% of activities), "fine" for silt and finer material (12.8% of activities), and "mixed" for mixtures of coarse and fine material (13.4% of activities). Material with no sediment type information (51.7% of all activities) was categorised as "mixed" (see Supplementary Table 1 and

**Table 3 | Numerical model configurations for the dredging and dumping simulations**

|  | Spin-Up | Dredging | Dumping |
|---|---|---|---|
| Model name | TOCMAIM | SCHISM-SED3D-TOCMAIM | SCHISM-SED3D |
| Model domain | North Sea | Northwest European shelf |  |
| Horizontal resolution | 0.03° | 4.5–15 km | 0.7–15 km |
| Simulation period (hydrographic forcing) | Jan 1st, 1950–Dec 31st, 2013 | Jan 1st–Dec 31st, 2000 | Jan 1st–Dec 31st, 2012 |
| Period of averaged dredging/dumping activity | – | 1992–2023 | 2014–2021 |
| Sediment classes | 3 organic carbon classes of different reactivities | 6 classes (clay, sand, silt, 3 organic carbon classes of different reactivities) | 1 class (representing silt or organic carbon) |
| References for setup | Zhang et al.[15,61] | Kossack et al.[66]; Porz et al.[24] | Chen et al.[67] |

The reader is referred to Supplementary Table 1 for detailed sediment model settings. The simulation periods are chosen to coincide with periods for which the respective model setups have been validated.

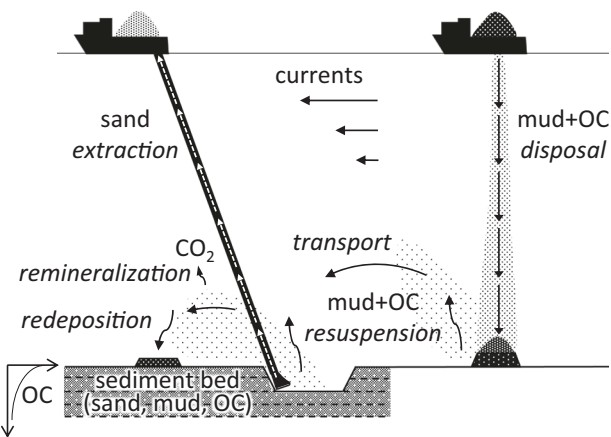

**Fig. 6 | Conceptual illustration of dredging and dumping processes in the model.** Dredging (or "extraction") removes the sand fraction of the sediment bed and resuspends organic carbon and fine-grained sediment (or "mud"). Dumping (or "disposal") places suspended material into the water column where it sinks to the bare seabed. In both cases, mud and organic carbon can be resuspended and transported by currents and redeposited elsewhere.

Supplementary Fig. 4). The "mixed" material was assumed to comprise coarse and fine sediment in equal parts.

Data on OC contents of dredged and dumped material is lacking. Therefore, a range of OC contents of 2–8% and 0–2% was assumed in the fine and coarse fraction, respectively, based on the composition of harbour muds sampled at the port of Hamburg[58]. Similar ranges for OC contents have been found in sediments of other dredged areas in the region, such as in the harbours of Dunkirk[59] and in the Scheldt Estuary[60].

**Numerical models**

We use a suite of process-based numerical models to simulate the impacts of aggregate extraction and the distribution of dumped material (Table 3 and Fig. 6).

The first model is the Total Organic Carbon Macrobenthos Interaction Model (TOCMAIM)[61], which accounts for OC reactivity through three sediment classes with different remineralisation rate constants, as well as lithogenic sediment dynamics. This model applies the phyto-detritus fields of the NPZD-type ecosystem model ECOSMO[62] to assign fresh input of OC at the seafloor, where it is consumed and mixed vertically by bioturbating macrobenthos and remineralised through microbial respiration. The model is run for 63 years, at which point a quasi-equilibrium between the OC reactivity classes has established. There is a strong seasonal signal of fresh (highly reactive) OC at the seafloor surface following the phytoplankton blooms, most of which is

remineralised by the end of the year. Therefore, we use the end-of year fields of the year 2013 and average the OC reactivity contents over the upper 30 cm, corresponding to the typical depth of furrows observed following trailer head suction dredging[63,64]. The outputs of this model are used to initialise the sediment and macrobenthos of a 3D coupled hydrodynamics and sediment transport model based on the SCHISM modelling system[65]. The validated setup covers the entire Northwest European shelf using an unstructured grid with a horizontal resolution of a few km in the North Sea[66].

For the simulation of material extraction, total extracted amounts were spread evenly across all extraction areas and implemented as a removal of the sand component along with corresponding additional resuspension of the remaining grain size classes (clay, silt and the three OC classes; see Supplementary Table 2 for details). Thereby, the simulated impact represents both the enhanced oxygen exposure due to resuspension and the erosion of the upper bed layer, exposing deeper layers to higher remineralisation rates. Using the hydrographic conditions of the year 2000, one full year of extraction is simulated for two scenarios: one with continual extraction throughout the entire year and another with extraction once per week for a full day (see Supplementary Methods 1 for details). The results are compared to a simulation of the same period without extraction.

For the dumping simulations, the spatial resolution of the SCHISM model grid was increased in the Wadden Sea and the German Bight in order to adequately represent the flow dynamics there[67]. The resolution in the German Bight ranges from 0.7 km to 5 km and is around 1 km in the Wadden Sea area with local refinements at complex features such as flow-confining channels. For a representative picture of recent activities, dumping was assumed to be continuous, and the dumping volumes of 2014–2021 were averaged over time. Based on the estimated dumped OC amounts, dumping was implemented as an additional constant source of water and fine-grained sediment tracer at each model grid cell nearest to each dumping location (see Supplementary Methods 2 for details). It is assumed that OC and fine-grained sediment (mud) behave similarly in the water column, since OC is typically attached to fine mineral grains in natural suspensions, and therefore the results represent both OC and mud dispersal.

The mineralisation rates of OC in harbour muds vary widely and cannot be feasibly estimated from the available data. As we are mainly interested in the general distribution of dumped OC and mud, we do not assign a degradation rate to the dumped material in the model, and instead treat it as an inert, mass-conservative tracer. As opposed to material extraction, the dumping of OC does not directly affect OC in the subsurface layers of the existing sediment bed. Therefore, no other grain size classes are set for this experiment.

Some dumping points are located close to the shore or within estuaries that are outside of the model domain. Where the distance from a dumping site to the nearest grid cell was more than 0.2°, the

dumping site was not considered in the model. This cutoff is chosen such that upstream and inshore dumping activities are discarded, while including nearshore or estuarine dumping activities that lay just outside the model domain. The disregarded dumping activities amount to 3.6% of the total dumped amount and are mostly located within UK estuaries, which are not well resolved in the model. The amount of excluded dumping in the Wadden Sea is only 0.3% of the total dumped amount.

The dynamic behaviour of dumped material is not well known. In the dumping model experiment, OC is treated as a sinking tracer with a constant sinking speed of $0.05\,\text{mm}\,\text{s}^{-1}$ (see Supplementary Table 3 for details). However, in natural suspensions, particles with high organic matter content form larger aggregates at high concentrations with significantly higher sinking velocities[68]. Therefore, a sensitivity experiment was carried out in which the dumped material was injected at the bottom model layer, directly above the seabed, instead of at the surface layer. This is motivated by observations and high-resolution simulations showing that dumped material initially collapses quickly at the seafloor[69]. As an alternative approach to account for the possible influence of aggregation, an additional simulation was carried out with a sinking rate increased by a factor of twenty. This high sinking speed of $1\,\text{mm}\,\text{s}^{-1}$ corresponds approximately to the maximum sinking speeds measured in the coastal southern North Sea[70]. We simulate one full year of dumping using the hydrographic conditions of 2012.

## Data availability
Extracted areas and amounts available at "EMODnet Human Activities, Aggregate Extraction" https://ows.emodnet-humanactivities.eu/geonetwork/srv/api/records/fde45abd-7bf3-4f05-869c-d1ce77f4ac63?language=all-4f05-869c-d1ce77f4ac63, "Effects of extraction of marine sediments on the marine environment 2005-2011" https://doi.org/10.17895/ices.pub.5498, and "Working Group on the Effects of Extraction of Marine Sediments on the Marine Ecosystem (WGEXT)" https://doi.org/10.17895/ices.pub.5733. Dumped amounts available at "OSPAR Dumping and Placement of Wastes or Other Matter at Sea" https://odims.ospar.org/en/submissions/ospar_dumping_at_sea_2021_01/. Modelled OC pools of different reactivities are available at "Field and Model Data for Bottom Trawling Impacts in the North Sea" https://doi.org/10.5281/zenodo.8297751. "Wadden Sea World Heritage Site" polygons available at http://marineregions.org/mrgid/26877. Source data for Fig. 2 and Fig. 3b are provided with this paper. Source data are provided with this paper.

## Code availability
The SCHISM model including the sediment module is available at https://doi.org/10.5281/zenodo.6537526. The TOCMAIM model is available at https://doi.org/10.17632/2vvny3xd85.2.

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

## Acknowledgements

This study is a contribution to the project APOC funded by the German Federal Ministry of Education and Research (BMBF) within the MARE:N programme (grant 03F0874C; L.P., J.C., W.Z., R.Y., and J.K.) and to the collaborative project KomSO (grant 3523NK370A-E; L.P., W.Z., J.K.) coordinated by the German Federal Agency for Nature Conservation (BfN). It is also supported by the Helmholtz research programme POF IV "The Changing Earth – Sustaining our Future" within "Topic 4: Coastal zones at a time of global change". This work used resources of the German Climate Computing Centre (DKRZ) granted by its Scientific Steering Committee (WLA) under project ID bg1244.

## Author contributions

Funding acquisition: W.Z., C.S., and L.P.; Technical implementation: L.P., J.C., and W.Z.; Data analysis: L.P., R.Y., and J.K.; Writing: L.P.; Editing: L.P., W.Z., R.Y., J.C., J.K., and C.S.

## Funding

## Competing interests

The authors declare no competing interests.
