## [Transparent Peer Review file · Nature Communications]

Dredging and dumping impact coastal fluxes of sediment and organic carbon

Corresponding Author: Dr Lucas Porz

Version 0:

Reviewer comments:

Reviewer #1

(Remarks to the Author)

The manuscript by Porz et al., provides a novel estimate of the impact of anthropogenic dredging and subsequent dumping on the natural flow of sediment and carbon within the coastal ocean. The authors modelling efforts demonstrate that sediment and carbon budgets need to include anthropogenic impacts, as the magnitude of these effects is of a similar magnitude to the natural fluxes. This manuscript fits well in a recent series of high impact papers that put the spotlight on the anthropogenic influence on coastal carbon cycling in a quantitative way.

The manuscript is well written and has important implications for our thinking of coastal dynamics, as well as for policy. However, the authors need to clarify how the impact on the OC remineralization is calculated and try and give a more nuanced picture. Furthermore, the discussion of the wider implications and relevance of these anthropogenic activities is a little immature and could do with a more thorough effort to contextualize these results in the broader picture of the growing blue economy and marine carbon dioxide removal efforts. I am confident the authors will be able to do address our comments within one revision, and I expect the final paper it will attract much attention from different scientific fields. I have summarized my other comments below.

Kind regards

Sebastiaan van de Velde & Luna Geerts

Major comments:

Much of the estimates regarding OC disturbances is based on 1 publication (29) which is specific for the Elbe estuary. The authors also mention on L74 that this harbor sludge is particularly rich in OC, which then puzzles me that this sediment is used to represent all the dumped sediment. I think more attention should go to quantifying this OC estimate by including other studies if possible. I am aware of at least one other paper that shows the organic matter of dredged sediment of the harbour of Dunkirk (<https://www.sciencedirect.com/science/article/abs/pii/S0956053X07001328>), and the Easter and Wester Scheldt are well studied systems.

Additionally, the OC estimates of these dumped sediments may be much smaller than expected as part of this dumped sediment was already dumped once before, at which point part of the OC may already have mineralized, reducing the total OC and increasing the fraction of refractory OC. The authors acknowledge this shortcoming (L140), still I would appreciate an effort in quantifying this effect. Since your simulations tell you how much sediment gets redeposited within the intertidal zone, you could make an estimate of how much material gets redeposited and correct for this – this would be a very relevant point to make, as it shows that much of the effort of dredging and dumping is pointless if the deposit zones are not chosen carefully. If this is not possible, I would hesitate to include these estimates in Fig. 3b as these OC inputs may be greatly overestimated. The same point holds for the sediment fluxes – your statement on L140 implies that sediment fluxes on a large scale (larger than estuary) are not impacted as much by anthropogenic impacts as most of the sediments are internally cycled. The manuscript would benefit from expanding on this point and doing an effort in quantifying this internal recycling.

A large implication of this study is that anthropogenic disturbances in OC and sediments are large and of a similar scale to natural sediment fluxes. However, I am missing quantifications of these natural sediment fluxes to which these anthropogenic disturbances are compared. For example, Table 1 would do well with an extra row showcasing the natural

volumes to put these numbers into context. Connected to this I am missing the significance of these results regarding the CO₂ impact. For example, how much CO₂ is potentially emitted because of these anthropogenic disturbances? How does this affect the CO₂ footprint of current coastal activities such as dredging and beach nourishment? These are valid points raised in the introduction but not discussed further.

Throughout the manuscript you use 'kt CO₂e y' (e.g. L102), it is unclear how these conversions are made. If it is emission-related (CO₂ to the atmosphere), does it come directly out of your model runs as a CO₂ flux? If not, did you take into account the buffering effect of the ocean carbonate system (Frankignoulle 1994), which means that only ~0.6 moles per mole of DIC added to the ocean will go to the atmosphere?

Or is it just a conversion of kt C to kt CO₂? In this case, it is misleading and you should not present it that way as it has been clear from the previous discussions in the literature that people tend to take the CO₂ numbers on face value as emissions, which they are not.

Please specify briefly in text or in the methods, and adjust accordingly

The different models used create a bit of confusion in how the results are interpreted. As I gather, your estimate of the impact of sand extraction is based on the lability of the organic carbon you simulated for an undisturbed system. So you assume that all the semi-labile and labile (as a sidenote, this are archaic denotations and should be replaced with reactive, unreactive, etc.) OC gets mineralized. But in your model run, the (semi-)labile fractions would be mineralized anyway over time. To make a realistic estimate, you need to estimate how much more OC would be broken down if disturbed versus how much would be if it remained in the sediment. You can do this by either running your complete model simulation, as you have done previously for mobile bottom-contact fishing, or by making other assumptions about the change in reactivity. But you cannot simply assume all organic carbon suddenly gets remineralized – this is the same mistake made previously in (Sala et al. 2021), and I hope the authors will rectify this. The sentence at L105 is a bit off-hand, and I don't think they are based on any real numbers – please update this as well.

For the dumping model, you essentially speculate again on OC reactivity, but you do not account for any uncertainty on the parameters. The same argument holds, if large estuaries are efficient bioreactors – would that OC then not be degraded without it being dredged and dumped? It is crucial to account for this 'what-if' scenario, otherwise we end up with inflated numbers again (see the discussion in (Hiddink et al. 2023) on the Sala estimates for mobile bottom-contact fishing). You have done this well in your previous manuscripts (Porz et al. 2024; Zhang et al. 2024), so I am a bit puzzled with how you have approached the current question. Please revise your estimates and try and account for the uncertainty in the assumptions.

Minor comments:

L11: That is not entirely correct, there are large-scale estimates of (for example) trawling, but there is a lot of uncertainty around these numbers. I would also immediately use 'organic carbon', as you are not dealing with the inorganic side of things.

L20-21 For clarity, I would rewrite "the amount being similar in magnitude..." to "which is similar (in magnitude) to..."

L28-29 Shorten the sentence, for readability. The sentence can be split up in two.

L34: again, not really phrased correctly given the recent series of papers of several groups (including yours) about mobile bottom-contact fishing. In particular the sediment fluxes have been discussed at length (see papers by - for example - Puig, Planques, and Paradis), while the impact on organic carbon has also received quite some attention (papers by Atwood, Hiddink, reviews by Tiano, Epstein).

L43 Remove "and" from "and harbours"

L45 Net sediment flux from what to where?

L47 Specify perhaps in brackets what the estimated riverine export is. Is it more or less?

L48-L49 Same here, specify.

L53 here you nuance what is understudied – but this should be done upfront

L55 Add references for publicly available data or refer to supplementary or methods where this is specified.

L65-L66 Sentence can be shortened for clarity instead maybe in short elaborate what the significance is of filtering out these fine particulates.

L76-L79 Sentence can be broken up so that subject remains clear. It is also unclear to me what "coastal human activities" in this sentence relates to, the element cycles? Or the pelagic ecosystem? I don't quite understand what the sentence is supposed to mean.

Figure 1. On a greyscale the extraction sites are very difficult to distinguish. Perhaps these polygons could be colored solid

like the circles? Or the line width could be altered? The difference between the isobath lines are also very hard to see.

L88-90 Please add reference.

L90-91 Typo/Rewrite to, for example; "In the year 2009 and 2010, the...".

L97 It was unclear to me where this value came from and on what it was based? Is it a value you derived? Or is a reference missing?

L98-102 Sentence can be broken up for clarity.

L106-107 Could you quantify up to how much % this refractory OC may remineralize?

Figure 2 & Figure 3b. Throughout the text the unit kt is used but in the figures, Mt is used instead. Maybe Mt can be used throughout the text for consistency?

Also, currently both Fig. 2 and Fig. 3 have two double y-axes which can be inherently confusing, especially when plotting two data series (e.g., Fig. 3b). Maybe the authors could instead consider creating a figure showcasing the total extraction rates (Fig. 2) without the OC, and supplement this with a subpanel figure showcasing the total sediment dumped (Fig. 3b) without the OC. Alternatively, if both figures can be combined in one figure that would be even better. Then the OC amounts from both extraction and dumping could then be combined into their own respective figure as well. This would aid in showing the reader which of the two processes adds the most OC and material. Additionally, it more clearly separates input data, from modeled results.

L110 Write OC in full for the figure captions.

L114-116 I miss a reference here. I would refer to the methods section where you discuss this data collection.

L120 and Figure 4. A reference to actual settling velocities or a mention about which types of sediments or grain sizes these settling velocities are valid for would help contextualize these velocities.

L125 I am confused, on how there is more OC in the water column in the case of the rapidly sinking OC compared to the slowly sinking OC. I would think that the slowly sinking OC remains more easily in suspension. Is this a mistake? If not, please address this as it is not apparent to me why this would be the case. Also, the use of "material" and OC is used interchangeably throughout the paragraph. Does "material" mean OC or sediment? Pick either OC or sediment as it was unclear to me whilst reading.

L128-130 What is the relevance of this? Maybe mention the CO₂ impact (L154) here already.

L131-132 does this total amount of mud deposition include also anthropogenic inputs? If not specify that this relates to natural mud deposition. If yes, then the conclusion (L133-134) would not be entirely correct. The natural sedimentation processes would be much smaller than anthropogenic inputs.

L140-142 I think this is very important point and deserves more attention either in the discussion or here (see the point made in the major comments section).

L149 Is it possible to give a lower limit as well? How big is the range?

Figure 3a is dumping in units "t C", is this the same as mass OC? If so, I would suggest adding this to OC axes as well (e.g., Fig. 3b), so that OC in dumped material is specified in units "Mt C".

Figure 4. I would place the legend from subplot (a) next to both figures as the legend is valid for both. Additionally, I would suggest making the legend entry for the Wadden Sea World Heritage Site slightly see through as in the figure itself. Also here I suggest specifying g C m⁻².

L175-177 At what time scales are these disturbance periods?

L180-184 I think this can be shortened considerably. I would also not directly convert this to a yearly estimate as this conversion makes little sense. The lifetime of a wind turbine has little in common with the repeated CO₂ impact of dredging and dumping to which you compare it to. The lifetime comparison only makes sense when analyzing the overall CO₂ impact of wind turbines (which you do later), but this is not what is being discussed here. I think it is fine, to state what the total impact is (400 kt CO₂), but that the overall CO₂ impact is smaller due to its lifetime (20 years, 50 ktC yr) and leave it at that.

L185 A similar positive or negative disturbance? Or in both ways?

L187 The effect of climate change on solute fluxes is heavily dependent on the region, no? Or do you mean specifically for the North Sea? If so specify.

Table 1 I would appreciate an extra row showcasing natural rates so that the magnitude of these disturbances can be compared to natural fluxes. Additionally, mention in the figure caption how the conversion from organic C to CO₂ is done.

L206 The natural magnitudes are not as obvious currently.

L212 Is it possible to estimate the magnitude of these effects?

L222 Magnitudes of what? Carbon? Sediment? Specify please.

L225 This statement deserves a little more elaboration – how exactly would alkalinity increase? I would instinctively say that any disturbances lead to alkalinity loss – not gain. I can see how harbour sediments that are enriched in fines and organic matter form more pyrite and release alkalinity (Hu and Cai 2011), but disturbances would lead to pyrite oxidation and alkalinity loss. Additionally, influshing of oversaturated water into the anoxic sediments could lead to more carbonate precipitation and subsequent alkalinity removal (2 papers have recently been accepted and published that could be used as more relevant references to alkalinity and disturbance events; van de Velde et al., Ocean alkalinity destruction by anthropogenic seafloor disturbances generates a hidden CO₂ emission, Science Advances, in press; Thanveer Kalapurakkal et al., Bottom trawling of muddy sediments enhances pyrite oxidation and carbon dioxide emissions, Communications Earth and Environment, in press)

L249 How many were these out of the total?

L262 I presume the correct reference is Table S1 not Table S2?

L263 Related with major comment. This seems like a critical assumption as all OC values are based on this one study. Can these values be supplemented with other studies? If data is lacking, please specify. Additionally, can you expect that using these OC values are valid for the entire model domain? If not do these estimates under or overestimate OC inputs?

L287 On what is this OC average based?

L292 Is the WC used here valid? Hoppers that dump sediment are not high in water content as this water is disposed of (L65-66). Should WC therefore not be much lower?

L296 This seems quite important, especially considering that much of this sediment will remain within the estuaries. As a result, you will overestimate the dumped sediment outside the estuaries as the total is averaged over the other dumping sites. Wouldn't it make more sense to reduce the total dumping amount by 20% to account for this or instead create dumping sites to grid cells nearest to the out of bound dumping sites?

Reviewer #2

(Remarks to the Author)

Reviewer #3

(Remarks to the Author)

Review Dredging and dumping impact coastal fluxes of sediment and organic carbon by Porz et al

The authors investigate the impact of dredging in the form of mineral aggregate extraction and dumping activities on the amounts of bulk sediments, organic carbon, remineralization with an emphasis on the Wadden Sea tidal flat system. They combine data analysis and process-based numerical modelling to conclude that dredging and dumping can impact the morphodynamics of coastal zones of the North Sea at the same magnitude as natural processes of erosion and sedimentation and then they recommend the consideration of the direct human impacts on sediment when assessing the sediment or carbon budget of coastal areas.

The objectives of the study are certainly relevant as the increased of coastal activities, notably dredging and dumping, affects the seafloor integrity, with modification of the sediment structure, water transparency and the cycling of essential elements like carbon and nitrogen with potential effect on carbon sequestration and the emission of greenhouse gasses. The authors have gathered data from various sources on dredging and dumping sites along the continental coast of the North Sea over the past three decades and incorporated them into their modelling setup, which combines several numerical models. This study complements the current understanding of the impact of offshore human activities on sedimentary OC and provides a quantitative regional-scale assessment of dredging and dumping impacts on sedimentary OC. The manuscript is quite well described but could benefit from English editing.

However, the manuscript does not present a significant advance in our holistic understanding of the effect of dredging and dumping activities on the coastal zone. The authors conclude that “ While this study gives first estimates of the magnitudes, further studies gauging impacts of sediment disturbing activities on the overall carbon cycle should include resolving indirect effects”. I fully agree with them. This study is important but is a starting point in the understanding of a complex process. For this reason and for others that I am detailing here below, the level of maturity of the methodology and the scope of the

conclusions are not mature enough to influence the thinking in this field. Then, I can not recommend publication of this manuscript in Nature communication and recommend that the authors submit their work in a specialist journal.

The authors used a model that combines an unstructured grid physical model, a NPZD-type biogeochemical model and a macrobenthos-organic carbon model. This model is used to (1) simulate the amounts and liabilities of organic carbon in the seabed to gauge the impacts of aggregate extraction and (2) simulate the distribution of dumped material.

The major criticisms are:

- 1) The mineral sediment is not part of this modelling framework although the authors correctly mention that the aggregate extraction includes a lot of sand and coarse sediment. The capacity of the model to answer the two above-mentioned questions depend on its ability to represent the deposition, erosion and resuspension process of Suspended Particulate Matter (SPM). This capability is not enough assessed in the papers to which the authors refer to. I would have liked to see a clear demonstration of the ability of the model to simulate the SPM, SPM plume and fluxes and, for instance, comparison with satellite data.
- 2) The impact of aggregate extraction is estimated as the flux of organic carbon out of the sediment at the dredging site. Then the fate of this extracted organic carbon is not investigated. It is possible that this OC is deposited at another place or that it will be degraded in the water column and then outfluxes to the atmosphere. The authors have a model able to estimate that.
- 3) The impact of dumping is assessed by estimating the place of deposition of OC (the mineral fraction is ignored as for dredging). The authors conclude that the deposition is important in the Estuary where small scale processes are important; The quality of the model in this region is not assessed. The authors refer to a paper by Kossack et al 2023 published in *Frontiers in marine research, biogeochemistry*. However, having read this *Frontiers* paper I see some differences mainly the *frontiers* paper focuses on larger scale. The authors correctly pointed out that the dynamics of sediment is really important in estuaries but the validation of the models in Estuaries where very small scales processes are important is not really addressed in the *Frontiers* papers,
- 4) There are a lot of uncertainties in model parameters like for instance the proportion of organic carbon, the liability of organic matter. These uncertainties are correctly pointed out by the authors but their impact on model estimations is not addressed. Considering the limitations of the validation exercise, a kind of ensemble simulations with different liability, OC fraction, sinking speed, mineral versus organic fraction, would have addressed these limitations. Same with the uncertainty in the distribution of extraction sites mentioned in section 4.1
- 5) Chapter 4.3 "Numerical models" is too brief and densely written given the importance of the information it contains, which makes it difficult to follow. The models are scarcely described, with the authors primarily referencing previously published articles, some dating back to the mid-2010s. Have there been any modifications to the model setups since then? A clearer explanation of how the models were used together, supported by a schematic figure and a brief description of the key features of the model configuration, would greatly improve reader comprehension. The implementation of dredging and dumping as modelled processes deserves separate subchapters.
- 6) A continuation of point 5: If I understand correctly (though it was unclear), the sediment model validation used in this study is presented in Chen, 2025 (ref. 58)? However, the validation in that paper focuses only on suspended particulate matter (SPM) dynamics in a small muddy area, which represents just a fraction of the Wadden Sea. How can we be confident that the model is capable of accurately simulating dumping in other areas of the North Sea, which are the focus of this study? The sediment bed texture across the North Sea is highly heterogeneous, and so are its mechanical properties—both of which influence bottom fluxes and therefore could significantly affect model results.
- 7) In the abstract, the authors state that they consider dredging and dumping to be important components of the global sediment and carbon budgets. Subchapter 2.3 is dedicated to comparing the impacts of various human activities typical of the North Sea. However, when comparing human-induced impacts with natural sedimentary cycles, the authors rely on estimates from the Wadden Sea as a reference—which represents only a small part of the North Sea. Can these findings be extended to the entire North Sea? And what about at a global scale?

Detailed Remarks:

Introduction

- 1) Lines 44–50: If the assessment of human activities is not directly compared to the natural marine sediment cycle (e.g., in absolute values or percentages), the significance of their impact is unclear. Riverine input is only one part of natural sediment dynamics; coastal erosion, atmospheric deposition, and other sources should also be considered.
- 2) Lines 49–50: "It is thus evident that any holistic analysis of contemporary matter fluxes in the Earth system must account for direct human impacts on sediment disturbance." — This statement may be premature in the Introduction, as it appears to be a conclusion derived later in the manuscript. Consider moving this sentence to the Discussion or Conclusion section where supporting evidence has already been presented.
- 3) Line 60: The recent report identifies dredging and dumping as primary risks to the Wadden Sea's World Heritage status. What exactly are these risks? How does offshore dumping of additional organic carbon impact benthic communities? It is suggested that these issues be discussed further in the Discussion section, especially since the Heritage status is only briefly mentioned elsewhere in the manuscript without clarification of what aspects are under threat or in need of protection.
- 4) Line 69: What role do benthic invertebrates play in this study? Their relevance is not immediately clear and could be explained more explicitly.

Figure 1 : The caption states: "Colour scheme shows modelled remineralization rate" — Is this output from the current study, or is it based on a previously published dataset or model? This should be clarified. Consider adding a zoomed-in panel showing the Wadden Sea and adjacent Southern Bight, if the figure is intended to highlight human activities in those regions. This would enhance readability and relevance.

Methodology

Section 4.3: See main points 1 & 2 in the general comments above.

Supplementary Materials

Table S2: Are the same parameter values (especially critical shear stress and settling velocity) applied to all three sediment classes defined in the model (sand, mixed, mud)? If not, please clarify how these parameters vary by class. If they are the same, this should be explicitly stated and briefly justified.

Results

- Lines 88–96: These paragraphs could be merged into one, as they cover related content and would benefit from improved flow.
- Line 97: The value 0.10% is given without a reference. Please provide a citation or clarify how this value was derived.
- Lines 99–100: The 87-10-3% partitioning of carbon types (refractory, semi-labile and labile) is introduced without explanation. Please elaborate on the rationale behind this specific partitioning—was it based on literature, model assumptions, or field measurements?
- Lines 99–100 & 104–105: There appears to be a contradiction between the statements: "Refractory carbon is unlikely to be remineralized following disturbance."

"Refractory carbon is not fully inert ... and may be made more bioavailable when exposed to oxygen and mixed with labile OC." These two points seem to conflict. Consider reconciling them by clarifying under what conditions some remineralization might occur, and whether it is considered significant in the context of your study.

- Line 107: The phrase "refractory OC remineralization should not exceed a few hundred kt CO₂e/yr" suggests a value that is an order of magnitude higher than your short-term remineralization estimates, yet it's presented with less emphasis. Please clarify how this value was calculated. What degradation rates in the water column were assumed for disturbed/resuspended refractory OC? It is also unclear whether these estimates are included in Table 1. Please indicate where they are documented or consider adding them.

Side question: After aggregate extraction, does the exposure of deeper, previously buried sediments to overlying ocean water accelerate OC degradation, for example through enhanced diffusive fluxes (e.g., oxygen penetration)? If so, it might be worth discussing briefly.

Line 114: A reference is missing here.

Lines 123–124: The statement regarding particle retention in the English Channel lacks supporting evidence. The Wilson et al. (2018) sediment maps show that the Dover Strait contains some of the coarsest sediments in the region (gravel fractions 10–70%), suggesting high energy and resuspension. This makes the claim that particles "remain in place after one year" difficult to accept without further clarification. If the area of interest is actually the mud-rich coastal zones of Belgium and the Netherlands, then this should be referred to as the Southern Bight, not the English Channel. Please clarify which area is meant.

Figure 4:

These maps are very interesting and merit more detailed discussion. Consider addressing the following:

What is the isobath that marks the edge of observable dumping impact on OC deposition?

Why is there a consistent white (zero-deposition) anomaly near the island of Texel across both maps?

How do the residual currents and tidal mixing unique to this area affect the final distribution of deposited OC?

If the settling velocity is assumed to be 1 mm/s, how does OC deposition occur in the offshore area between the UK and the Netherlands, which lies tens of kilometers from the nearest dumping site?

What is the baseline distribution of natural mud deposition in the southeastern North Sea? How does the spatial footprint of dumping-related OC deposition compare to natural OC deposition?

Which areas are most affected by dumping, and why? Please consider a more explicit spatial comparison in the text or with an additional figure.

Chapter 2.3 & Table 1:

Estimates for natural sedimentary and OC disturbances and OC remineralization are not clearly integrated into the comparison with anthropogenic impacts. This connects directly to major remark 3 and should be addressed to strengthen the broader relevance of your impact assessment.

Discussion

Line 206: "We show that dredging and dumping can impact the morphodynamics of coastal zones at the same magnitude as natural processes of erosion/sedimentation." — This is a strong claim that is not fully demonstrated in the results. It is only briefly mentioned (see comments on Figure 4). Consider supporting this with a dedicated figure or map showing annual OC deposition from human activities (e.g., dumping) as a percentage of natural OC deposition, with dumping sites overlaid.

General: Are there any recommendations for regional dumping site management based on your findings? For example, the result that 22% of dumped mud eventually returns to the same river channels it was dredged from suggests a potentially large inefficiency. This is a striking and policy-relevant insight but is not currently mentioned in the Discussion. Highlighting this could add practical impact to your study.

Minor comments

Define blue carbon ecosystems

"the global net sediment flux is due to one or several of these activities" what is meant here? What is the global net sediment

flux?

Line 102: please define kt CO₂e/yr

Table 1 : Please explain why OWF does not disturb the sediment

Nutrients in resuspension impact on biology same with the river. Please clarify.

Line 97: The average concentration of sediment OC at the extraction sites is 0.10% and from the estimation of sand and gravel estimate the amount of organic carbon extracted. First, this 0.1 % seems to be highly empirical and is poorly validated

Lines 103-105: For comparison, the recent fuel emissions of the European dredging fleet was estimated at 600-800 kt CO₂e/yr according to an industry report. Nevertheless, it is important to note that the refractory OC fraction is not fully inert, and may be made more bioavailable when exposed to oxygen, especially when mixed with more labile compounds. This is not clear how the authors have taken this uncertainty into account.

Line 116-117: carbon contained in dumped material is in the range of 500-5,000 kt/yr, reflecting the uncertainty in the OC content of dumped material. How does this uncertainty be taken into account?

Reviewer #4

(Remarks to the Author)

Reviewer #5

(Remarks to the Author)

The manuscript "Dredging and dumping impact coastal fluxes of sediment and organic carbon" by Porz and coauthors builds on earlier work by the team to assess turnover of carbon on the continental shelf of the North Sea by anthropogenic activities. In Zhang et al 2024 (<https://doi.org/10.1038/s41561-024-01581-4>), the authors evaluated the impact of bottom trawling on carbon remineralization and release to the atmosphere. Using a similar approach and modeling environment, in this study the authors evaluate the carbon consequences from aggregate extraction and material dumping. This is an interesting modeling study and below I highlight questions and places to strengthen the manuscript:

Abstract: It is a bit disingenuous to start the abstract noting the lack of large scale estimates of the impact of anthropogenic activities on coastal sediment carbon dynamics given the spate of recent articles published, including by this author team.

[e.g. <https://doi.org/10.1038/s41586-023-06014-7>, <https://doi.org/10.5194/bg-21-2547-2024>, 2024, <https://doi.org/10.3389/fmars.2023.1125137>, <https://doi.org/10.1016/j.wasman.2021.11.031>, and others]

Introduction: Figure 1 is introduced here with mineralization rates shown. Are these rates attributed to natural processes plus anthropogenic disruption due to extraction and dumping or are these background rates from the model without those impacts? If these rates are part of the findings of the paper, it would be odd to only mention Figure 1 in the Introduction.

Results: Better referencing and context is needed to support the amount of carbon available to release to the atmosphere. Comparison with in situ measured rates of benthic oxygen demand/utilization would be useful as well.

"The average concentration of sediment OC at the extraction sites is 0.10%." Reference?

"In our model, 87% of this disturbed OC constitutes the refractory fraction, which is unlikely to be remineralized following disturbance, while 10.0% and 3.0% of OC are semi-labile and labile..." Reference?

"Nevertheless, it is important to note that the refractory OC fraction is not fully inert, and may be made more bioavailable when exposed to oxygen, especially when mixed with more labile compounds³⁴" What is the depth of oxygen penetration considered in the remineralization model? What is the depth of sediment typically extracted? Typically sandy sediments have advective transport and mixing down to ~15 cm, resulting in regular exposure to oxygen.

[<https://doi.org/10.1016/j.earscirev.2022.103987>]

Results: Given the potential variability in some of these model parameters, it would be useful to have a deeper discussion about sources of uncertainty and present more broadly a sensitivity analysis and potential impact on values in Table 1. It is good those values are order of magnitude, but within the context of the manuscript it would be good to understand how certain they are and how sensitive they are to some of the model assumptions.

Discussion: "For example, metabolic processes may generate 16 225 not only dissolved inorganic carbon, but also alkalinity, which under some circumstances can strengthen the coasts role as a carbon sinks⁴⁷." The impact of changes in organic matter remineralization on organic and inorganic carbon and alkalinity fluxes is definitely important. Alkalinity is typically generated during anoxic remineralization reactions (see <https://linkinghub.elsevier.com/retrieve/pii/B9780323997621000322> for a good review). What mechanisms do the authors think would lead to increased anoxia in bottom sediments with extraction and dumping compared to no disturbance conditions? A recently published paper indicated that enhanced oxygenation may lead to pyrite oxidation, resulting in loss of alkalinity and acidification enhancement [<https://www.science.org/doi/10.1126/sciadv.adp9112>], counter to the production invoked in this manuscript.

Version 1:

Reviewer comments:

Reviewer #1

(Remarks to the Author)

I have read the new version of the manuscript, and I am happy with how the authors have addressed all comments. I have a few small suggestions for improving clarity, but I think this manuscript is a great addition to the scientific literature and will attract a lot of attention from the scientific and policy-making communities.

Kind regards

Sebastiaan van de Velde

L78: define 'aqueous CO₂ emissions' for non-specialists

L84: 21-65% relates to the total mineralisation rate or the mineralisation of the unreactive fraction alone? If it is the unreactive fraction, you would have to first get a rate for the unreactive fraction, and then calculate the increase – not just multiply the whole rate by 1.65.

In any case, it is a bit unclear if phrased this way.

L224: I know I mentioned 0.6 in relation to the buffering capacity of seawater – but that was a mistake on my part – 0.6-0.7 related to the CO₂ release for calcium carbonate precipitation. In perfect equilibrium with the atmosphere, and assuming the pCO₂ in the atmosphere does not change, a DIC added to the water-column will always degass, since the alkalinity and pCO₂ set the DIC concentration. The 0.6-0.7 relates to the fact that not all water will re-equilibrate with the atmosphere due to circulation, and is thus a global number – which is not the same as the buffering capacity.

L238: I am not sure what riparian means, and I suspect I am not alone – maybe replace with a more common word?

Reviewer #2

(Remarks to the Author)

Reviewer #3

(Remarks to the Author)

The revised article has improved in quality, but I argue that with this advanced model and the extensive and detailed databases for dredging and dumping, the manuscript needs to expand the analysis of simulation results and balance it between both the Wadden Sea and the Southern Bight as two distinct regions. Model runs with two different sinking speeds of particles are not enough for me to believe that this process is as important for carbon disturbance as rivers, even on the scale of the Wadden Sea. I understand the challenges coming from quantification and parameterization of processes as dredging and dumping, but with a model like that their impacts should be analyzed deeper for the purpose of transferability of model results:

Major questions:

1. You model the entire North Sea, but the results disproportionately focus on the Wadden Sea, while the Southern Bight, which is equally busy in terms of dumping/dredging, is almost ignored or counted as a part of the greater North Sea (e.g. "Wadden Sea" or "German Bight" are mentioned 24 times in the manuscript, "Southern Bight" and "Southern North Sea" (which I believe also includes the Wadden) - less than 10). What are the differences between the Wadden Sea and SBNS for the fate of dumped carbon? How hydrodynamics, composition of the sediment bed and local biogeochemistry shape the fate of the dumped matter? Can Wadden sea depocenter absorb some scattered deposited material?
2. The flocculation process, that is disproportionately important for dumping, is implemented in the model with a sinking speed x20 than a real one. What is the source of this value? Regardless, I would either create another class of flocculated mud and define flocculation intensity as a function of mud content in the water, or, alternatively, directly parameterize the sinking speed as this function. It will allow a more realistic spread.
3. In Supplement: "For the dumping experiments, the seafloor is left bare, i.e., no sediment is initialized in the model". Does your model account for different properties of sand and mud in terms of deposited carbon retention (or it's only the shear stress)? Does your model (in this dumping experiment) simulate the natural carbon cycle (it was not clear for me)? Bioturbation/bioirrigation brings carbon into deeper layers, but if naturally deposited carbon is absent, irrigation will exclusively act on the dumped material, overestimating its degradation and burial.
4. It would be really interesting to see a relative impact (%) of dumping compared to the natural deposition as it is in the North Sea, with a significance horizon of 5%. It will show how localized the impact of dumping is, regardless of its absolute contribution to the carbon budgets on the regional scale. Same stands for dredging.

5. I thought about sensitivity tests for different proportions of sediments and carbon in the dumped material, but you have created a long realistic database. In this case, how are the sites where OC-rich mud is dumped, different from OC-poor sandy sediment sites? I believe that if we go into details of dumping materials, an analysis or elaboration should be done about impacts of those different materials.

6. Methodology: it needs a comprehensive sketch of the processes, represented in the model, with some additional info about sediment classes, organic carbon content range etc. As a reader, I had a hard time switching constantly between explanation in methodology chapter, supplement and numerous references to the previous setups to understand the modelling logic of the setup.

7. Figure 1 (minor): Zoom-in is still needed for the relevant regions: the SBNS and the Wadden Sea to demonstrate the sites. I didn't understand your comment about the Wadden sea and vulnerability map.

The current analysis for the North Sea is quite region-specific thus limiting manuscript's relevance to mostly the North Sea. Expanding the analysis will help a reader from any part of the world to project the findings on their region of interest and get first-hand estimates without a need to run a coupled model. Therefore I believe, addressing this criticism will align the manuscript with the standards of Nature.

Recommended: major revision.

Reviewer #4

(Remarks to the Author)

The authors provided detailed and substantive updates to their manuscript in response to the 5 reviews. These changes have resulted in a clearer and more robust product. All of my concerns and questions have been addressed.

Version 2:

Reviewer comments:

Reviewer #3

(Remarks to the Author)

MINOR NOTES

I think the manuscript is well-written and all my notes were taken into account. I just remind to be aware, that the SBNS is not as non-depositional as it might appear in some setups. Of course it's not the Trench, but De Borger's campaign have shown OC content in the sediment bed. Some of the PP is deposited during the leap tide and is not resuspended back.

I also highlight very good schematics and images. I think the authors have done a great work on understanding the processes of dredging and dumping and I recommend the manuscript for the publication.

Please consider the notes down below as suggestions to improve the manuscript.

MAIN TEXT

Add explicit statement that dumping material is inert, no OC remineralisation included.

Add justification why dumping uses only one tracer class while extraction uses six.

Insert mechanistic hydrodynamic contrast between Wadden Sea and Southern Bight.

Add fine = OC-bearing explanation in dumping introduction.

Add global assumptions link (OC content) to Methods.

Clarify circular dredging does not bias spatial patterns.

Add note on log-scale inflation in Fig. 4 caption.

Add tidal basin definition reference near Table 1.

Add hydrographic year mismatch explanation (2000 vs 2012).

Add 0.2° exclusion rule explanation to main Methods.

SUPPLEMENT

Clarify fine/coarse OC roles in Table 1.

Add "bare seafloor" explanation.

Justify 0.2° cutoff.

Clarify $C = 296 \text{ kg/m}^3$ derivation.

Add rationale for 0.05 and 1 mm/s.

Add log-scale note for Supplement Fig. 2.

Add explanation for Dec 30, 2014 shear stress date.

Link Supplement Fig. 4 to OC-bearing logic.

FIGURES

Add missing Wadden Sea blank-vulnerability note to Fig. 1 caption.

Add midpoint OC-range information to Fig. 3a/b captions.

Reviewers #1+2 comments	Author responses
This manuscript fits well in a recent series of high impact papers that put the spotlight on the anthropogenic influence on coastal carbon cycling in a quantitative way. The manuscript is well written and has important implications for our thinking of coastal dynamics, as well as for policy. However, the authors need to clarify how the impact on the OC remineralization is calculated and try and give a more nuanced picture. Furthermore, the discussion of the wider implications and relevance of these anthropogenic activities is a little immature and could do with a more thorough effort to contextualize these results in the broader picture of the growing blue economy and marine carbon dioxide removal efforts. I am confident the authors will be able to do address our comments within one revision, and I expect the final paper it will attract much attention from different scientific fields. I have summarized my other comments below.	We thank both reviewers for their insightful comments that have helped us greatly in improving the quality of our manuscript. We have addressed all comments in our revised manuscript and provide point-to-point responses below. Responses indicating changes to the manuscript are indicated in italics.
Much of the estimates regarding OC disturbances is based on 1 publication (29) which is specific for the Elbe estuary. The authors also mention on L74 that this harbor sludge is particularly rich in OC, which then puzzles me that this sediment is used to represent all the dumped sediment. I think more attention should go to quantifying this OC estimate by including other studies if possible. I am aware of at least one other paper that shows the organic matter of dredged sediment of the harbour of Dunkirk (https://www.sciencedirect.com/science/article/abs/pii/S0956053X07001328), and the Easter and Wester Scheldt are well studied systems.	Thank you for the suggestion. Unfortunately, little public data exists for the OC contents of dredged harbour sediments specifically. We have added reference to Verlaan (2000) for Scheldt estuary and Siham et al. (2008) for the harbour of Dunkirk, which support our assumptions about the average OC contents and their uncertainties.
Additionally, the OC estimates of these dumped sediments may be much smaller than expected as part of this dumped sediment was already dumped once before, at which point part of the OC may already have mineralized, reducing the total OC and increasing the fraction of refractory OC. The authors acknowledge this shortcoming (L140), still I would appreciate an effort in quantifying this effect. Since your simulations tell you how much sediment gets redeposited within the intertidal zone, you could make an estimate of how much material gets redeposited and correct for this – this would be a very relevant point to make, as it shows that much of	It is true that the numbers on dumping will be inflated due to it being partly the same material, and that this complicates estimates of carbon fluxes. Since re-dredging of material redeposited in the nearshore occurs locally and our model does not resolve harbour basins or shipping channels well, estimating the amount that is “double-billed” is somewhat speculative. Based on the amount deposited in the nearshore after one year in our model, it may be in the range of 25-40% and we’ve now added this quantity to the text.

the effort of dredging and dumping is pointless if the deposit zones are not chosen carefully. If this is not possible, I would hesitate to include these estimates in Fig. 3b as these OC inputs may be greatly overestimated. The same point holds for the sediment fluxes – your statement on L140 implies that sediment fluxes on a large scale (larger than estuary) are not impacted as much by anthropogenic impacts as most of the sediments are internally cycled. The manuscript would benefit from expanding on this point and doing an effort in quantifying this internal recycling.	Note that although not well documented in the scientific literature, this phenomenon is known in harbour management, at least for the Elbe estuary (“Kreislaufbaggerei”; “circular dredging”), but it can be much more expensive and/or illegal to dump material farther offshore where it is less likely to return; Only a limited amount of dredged material is dumped offshore. We took it more as confirmation of our model that this tidal pumping phenomenon is reproduced in our simulations, and we’ve now clarified this in the text.
A large implication of this study is that anthropogenic disturbances in OC and sediments are large and of a similar scale to natural sediment fluxes. However, I am missing quantifications of these natural sediment fluxes to which these anthropogenic disturbances are compared. For example, Table 1 would do well with an extra row showcasing the natural volumes to put these numbers into context. Connected to this I am missing the significance of these results regarding the CO2 impact. For example, how much CO2 is potentially emitted because of these anthropogenic disturbances? How does this affect the CO2 footprint of current coastal activities such as dredging and beach nourishment? These are valid points raised in the introduction but not discussed further.	We have added rows with riverine fluxes, atmospheric deposition and marine inflow/outflow and natural sediment deposition based on literature to Table 1. The CO2-values are mentioned in relation to emissions from the mineral extraction fleet. We’ve added a part on potential atmospheric emissions to the discussion.
Throughout the manuscript you use ‘kt CO2e y’ (e.g. L102), it is unclear how these conversions are made. If it is emission-related (CO2 to the atmosphere), does it come directly out of your model runs as a CO2 flux? If not, did you take into account the buffering effect of the ocean carbonate system (Frankignoulle 1994), which means that only ~0.6 moles per mole of DIC added to the ocean will go to the atmosphere? Or is it just a conversion of kt C to kt CO2? In this case, it is misleading and you should not present it that way as it has been clear from the previous discussions in the literature that people tend to take the CO2 numbers on face value as emissions, which they are not. Please specify briefly in text or in the methods, and adjust accordingly	OC loss was converted to CO2 without accounting for buffering, and we agree that this can be misleading. We now removed the conversion to CO2 in Table 1 and clarify throughout the manuscript, wherever CO2 is mentioned, that these are aqueous emissions, and also mention the buffering effect in the Discussion.
The different models used create a bit of confusion in how the results are interpreted. As I gather, your	We agree with the reviewer and have performed additional simulations modelling the disturbance by

estimate of the impact of sand extraction is based on the lability of the organic carbon you simulated for an undisturbed system. So you assume that all the semi-labile and labile (as a sidenote, this are archaic denotations and should be replaced with reactive, unreactive, etc.) OC gets mineralized. But in your model run, the (semi-)labile fractions would be mineralized anyway over time. To make a realistic estimate, you need to estimate how much more OC would be broken down if disturbed versus how much would be if it remained in the sediment. You can do this by either running your complete model simulation, as you have done previously for mobile bottom-contact fishing, or by making other assumptions about the change in reactivity. But you cannot simply assume all organic carbon suddenly gets remineralized – this is the same mistake made previously in (Sala et al. 2021), and I hope the authors will rectify this.	aggregate extraction, which are then compared to the undisturbed simulation and the numbers have been adjusted accordingly. We have replaced “labile”, “semi-labile” and “refractory” with “high reactivity”, “medium reactivity” and “low reactivity”, respectively, throughout the manuscript.
The sentence at L105 is a bit off-hand, and I don’t think they are based on any real numbers – please update this as well.	Quantification of the priming effect added.
For the dumping model, you essentially speculate again on OC reactivity, but you do not account for any uncertainty on the parameters. The same argument holds, if large estuaries are efficient bioreactors – would that OC then not be degraded without it being dredged and dumped? It is crucial to account for this ‘what-if’ scenario, otherwise we end up with inflated numbers again (see the discussion in (Hiddink et al. 2023) on the Sala estimates for mobile bottom-contact fishing). You have done this well in your previous manuscripts (Porz et al. 2024; Zhang et al. 2024), so I am a bit puzzled with how you have approached the current question. Please revise your estimates and try and account for the uncertainty in the assumptions.	For the dumping model, we do not prescribe any reactivity, since we used a conservative tracer to represent fine-grained sediment dispersal from dumping sites. We do agree that remineralization should always be compared to an undisturbed reference, and our model cannot resolve the undisturbed remineralization in harbours and shipping channels due to insufficient model grid resolution and lack of consideration of relevant biogeochemical processes. On the other hand, since harbours and shipping channels are artificial structures to begin with that are designed to be regularly dredged to maintain their functioning, the “undisturbed” state of a harbour is undefined. This has been clarified in the discussion and the excess remineralization for dumping removed from Table 1, focusing instead on the dumping impacts on lateral OC fluxes, which our model can resolve.
L11: That is not entirely correct, there are large-scale estimates of (for example) trawling, but there is a lot of uncertainty around these numbers. I would also immediately use ‘organic carbon’, as you are not dealing with the inorganic side of things.	Agreed. Rephrased accordingly.
L20-21 For clarity, I would rewrite “the amount being similar in magnitude...” to “which is similar (in magnitude) to...”	Done

L28-29 Shorten the sentence, for readability. The sentence can be split up in two.	Done
L34: again, not really phrased correctly given the recent series of papers of several groups (including yours) about mobile bottom-contact fishing. In particular the sediment fluxes have been discussed at length (see papers by - for example - Puig, Planaques, and Paradis), while the impact on organic carbon has also received quite some attention (papers by Atwood, Hiddink, reviews by Tiano, Epstein).	Agreed, we have added a sentence to acknowledge some of the work on bottom-contacting fisheries. (Note that we unfortunately cannot cite too much more of the work done in this area since we do need to limit the total number of references.)
L43 Remove “and” from “and harbours”	The sentence has been deleted as part of restructuring.
L45 Net sediment flux from what to where?	It was a general statement, since the sources and sinks can be different in each system, but we added “lateral” for clarification.
L47 Specify perhaps in brackets what the estimated riverine export is. Is it more or less?	Added
L48-L49 Same here, specify.	Added
L53 here you nuance what is understudied – but this should be done upfront	We have reorganized the introduction to mention dredging and dumping earlier.
L55 Add references for publicly available data or refer to supplementary or methods where this is specified.	Referred to methods section
L65-L66 Sentence can be shortened for clarity instead maybe in short elaborate what the significance is of filtering out these fine particulates.	Rephrased; this part has been moved to Methods to streamline the Introduction
L76-L79 Sentence can be broken up so that subject remains clear. It is also unclear to me what “coastal human activities” in this sentence relates to, the element cycles? Or the pelagic ecosystem? I don’t quite understand what the sentence is supposed to mean.	Rephrased
Figure 1. On a greyscale the extraction sites are very difficult to distinguish. Perhaps these polygons could be colored solid like the circles? Or the line width could be altered? The difference between the isobath lines are also very hard to see.	We understand the reviewers’ concerns, but we believe this may be due to the quality of the preprint figures. We have checked the figure in greyscale and are able to clearly distinguish the extraction polygons since they already have a greater line width. Therefore, we prefer to keep these polygons semi-transparent as it allows a sense of the underlying carbon liability. The isobaths can also be distinguished in the full-quality image. We therefore would like to keep the image this way but will take these comments into account in case the editorial office has similar concerns.
L88-90 Please add reference.	These numbers are the result of the data analysis carried out in the study. All references are given in the Methods so we’d rather not repeat them here, but

	have re-worded for clarification.
L90-91 Typo/Rewrite to, for example; “In the year 2009 and 2010, the...”.	Done
L97 It was unclear to me where this value came from and on what it was based? Is it a value you derived? Or is a reference missing?	This is based on sediment OC map of Bockelmann et al. (2018). Added
L98-102 Sentence can be broken up for clarity.	Done
L106-107 Could you quantify up to how much % this refractory OC may remineralize?	Yes, this has been added with our new simulation results.
Figure 2 & Figure 3b. Throughout the text the unit kt is used but in the figures, Mt is used instead. Maybe Mt can be used throughout the text for consistency?	We now use kt everywhere for consistency, since we think it makes most of the numbers more readable.
Also, currently both Fig. 2 and Fig. 3 have two double y-axes which can be inherently confusing, especially when plotting two data series (e.g., Fig 3b). Maybe the authors could instead consider creating a figure showcasing the total extraction rates (Fig. 2) without the OC, and supplement this with a subpanel figure showcasing the total sediment dumped (Fig. 3b) without the OC. Alternatively, if both figures can be combined in one figure that would be even better. Then the OC amounts from both extraction and dumping could then be combined into their own respective figure as well. This would aid in showing the reader which of the two processes adds the most OC and material. Additionally, it more clearly separates input data, from modeled results.	This is an interesting idea, but we came across some difficulties when plotting the data as suggested by the reviewers; since the magnitude of disturbed OC is so different between extraction and dumping, the extraction would hardly be visible in a combined plot; especially the individual lability fractions could no longer be distinguished. Another issue is that both data cover slightly different time ranges. We therefore prefer to keep the figures separate.
L110 Write OC in full for the figure captions.	Done for all figure captions
L114-116 I miss a reference here. I would refer to the methods section where you discuss this data collection.	Rephrased as “according to our analysis” instead of “is reported as” and added reference to Methods.
L120 and Figure 4. A reference to actual settling velocities or a mention about which types of sediments or grain sizes these settling velocities are valid for would help contextualize these velocities.	Added
L125 I am confused, on how there is more OC in the water column in the case of the rapidly sinking OC compared to the slowly sinking OC. I would think that the slowly sinking OC remains more easily in suspension. Is this a mistake? If not, please address this as it is not apparent to me why this would be the case. Also, the use of “material” and OC is used interchangeably throughout the paragraph. Does “material” mean OC or sediment? Pick either OC or sediment as it was unclear to me whilst reading.	You are right, we have switched those numbers. Thank you for finding this error. Corrected and checked that the other numbers are correct. We have changed “material” to “fine-grained sediment”; since the dumped tracer corresponds to fine-grained sediment class.
L128-130 What is the relevance of this? Maybe mention the CO2 impact (L154) here already.	The relevance is mainly that dumped material may contribute to the observed growth of the tidal flats as they keep pace with sea level rise. Added and

	removed CO2-relevance as part of restructuring.
L131-132 does this total amount of mud deposition include also anthropogenic inputs? If not specify that this relates to natural mud deposition. If yes, then the conclusion (L133-134) would not be entirely correct. The natural sedimentation processes would be much smaller than anthropogenic inputs.	The cited study compiled a mud budget for the entire system, which included the dredging sites, so the internal redistribution by dumping and dredging was not considered in that budget, except for the small amount of dredged material disposed of on land. In other words, about 8,000 kt/yr are imported to the Wadden Sea by rivers and from the North Sea, and we found that about the same amount is internally redistributed by dredging and dumping. We hope our edits now have this clarified.
L140-142 I think this is very important point and deserves more attention either in the discussion or here (see the point made in the major comments section).	As commented above, this is a known phenomenon among coastal management. However, we agree that it may not be as established among scientists and the public, so we've expanded that point.
L149 Is it possible to give a lower limit as well? How big is the range?	We have calculated the range based on lower limits for labile OC found by Zander et al. (2020) and now the full range of values is provided.
Figure 3a is dumping in units "t C", is this the same as mass OC? If so, I would suggest adding this to OC axes as well (e.g., Fig3b), so that OC in dumped material is specified in units "Mt C".	We have changed both axes' labels to be consistent. We chose ktC for the whole manuscript since we think it makes most of the numbers more readable.
Figure 4. I would place the legend from subplot (a) next to both figures as the legend is valid for both. Additionally, I would suggest making the legend entry for the Wadden Sea World Heritage Site slightly see through as in the figure itself. Also here I suggest specifying g C m ⁻² .	Done
L175-177 At what time scales are these disturbance periods?	Added
L180-184 I think this can be shortened considerably. I would also not directly convert this to a yearly estimate as this conversion makes little sense. The lifetime of a wind turbine has little in common with the repeated CO2 impact of dredging and dumping to which you compare it to. The lifetime comparison only makes sense when analyzing the overall CO2 impact of wind turbines (which you do later), but this is not what is being discussed here. I think it is fine, to state what the total impact is (400 kt CO2), but that the overall CO2 impact is smaller due to its lifetime (20 years, 50 ktC yr) and leave it at that.	Agreed, this has been shortened.
.L185 A similar positive or negative disturbance? Or in both ways?	Disturbance leading to OC removal. This should be clearer now that the previous sentence is removed.
L187 The effect of climate change on solute fluxes is heavily dependent on the region, no? Or do you mean specifically for the North Sea? If so specify.	Agreed, we removed climate change, since the general trend of increased solute fluxes and decreased particulate fluxes holds for most regions globally, and the role of climate change is not as clear.

Table 1 I would appreciate an extra row showcasing natural rates so that the magnitude of these disturbances can be compared to natural fluxes. Additionally, mention in the figure caption how the conversion from organic C to CO ₂ is done.	We have added rows with natural flux estimates and removed CO₂ in favour of “remineralized carbon” in ktC.
L206 The natural magnitudes are not as obvious currently.	We hope the changes mentioned above are able to clarify this.
L212 Is it possible to estimate the magnitude of these effects?	At the current stage, we believe there is insufficient data to quantify the magnitudes of these effects in a meaningful manner, so we would not like to speculate here but leave it as a suggestion for future research.
L222 Magnitudes of what? Carbon? Sediment? Specify please.	Rephrased
L225 This statement deserves a little more elaboration – how exactly would alkalinity increase? I would instinctively say that any disturbances lead to alkalinity loss – not gain. I can see how harbour sediments that are enriched in fines and organic matter form more pyrite and release alkalinity (Hu and Cai 2011), but disturbances would lead to pyrite oxidation and alkalinity loss. Additionally, influshing of oversaturated water into the anoxic sediments could lead to more carbonate precipitation and subsequent alkalinity removal (2 papers have recently been accepted and published that could be used as more relevant references to alkalinity and disturbance events; van de Velde et al., Ocean alkalinity destruction by anthropogenic seafloor disturbances generates a hidden CO ₂ emission, Science Advances, in press; Thanveer Kalapurakkal et al., Bottom trawling of muddy sediments enhances pyrite oxidation and carbon dioxide emissions, Communications Earth and Environment, in press)	This was in reference to ports being sources of alkalinity, but we agree that this is not specific to sediment disturbance, so we removed the corresponding reference and added the other two on pyrite oxidation.
L249 How many were these out of the total?	We have removed this sentence since the activities without site codes are listed later in the paragraph along with omitted amounts.
L262 I presume the correct reference is Table S1 not Table S2?	Yes, corrected.
L263 Related with major comment. This seems like a critical assumption as all OC values are based on this one study. Can these values be supplemented with other studies? If data is lacking, please specify. Additionally, can you expect that using these OC values are valid for the entire model domain? If not do these estimates under or overestimate OC inputs?	It is true that there is a lack of data on OC content in dumped material. This has been added. Evidently, the OC content can vary quite a bit even within a single estuary. However, we consider the total range of 0 to 8% to be quite large and should cover the possible range in most harbours and estuaries, which is reflected in the large uncertainty of OC disturbance in the results. Added reference to studies of Dunkirk harbour and the

	Scheldt Estuary. We've also added Supplementary Figure 2 showing the model results for min and max of OC contents in dumped material.
L287 On what is this OC average based?	This is the midpoint of the estimated range of 2-8% in fine-grained sediment based on the literature (see response above). Now clarified. Actually, the smaller OC amount contained in the coarse-grained fraction is also considered, so this was also added.
L292 Is the WC used here valid? Hoppers that dump sediment are not high in water content as this water is disposed of (L65-66). Should WC therefore not be much lower?	It is true that the value for WC in dumping may be overestimated in our model. However, we argue that does not affect the validity of our simulations since this would only change the (negligible) amount of water added along with the dumped sediment, and the important part is the total amount of sediment added. The purpose of adding water is to avoid extreme concentrations that would become a problem for the numerical schemes. We have added this explanation to the description in Supplementary Methods 2. Since this rather technical description might not be relevant for many readers, this portion has been moved to the Supplementary Information along with details on the newly added material extraction simulation setup.
L296 This seems quite important, especially considering that much of this sediment will remain within the estuaries. As a result, you will overestimate the dumped sediment outside the estuaries as the total is averaged over the other dumping sites. Wouldn't it make more sense to reduce the total dumping amount by 20% to account for this or instead create dumping sites to grid cells nearest to the out of bound dumping sites?	This is a good point, we have re-checked our calculations on how these inputs were generated. Most of those "missing" dumping points are located in UK estuaries, which are not as well resolved in our model, and only 0.3% of dumping within the Wadden Sea is not accounted for in this way. Since we only analyse the model results for the Wadden Sea in detail, those should not be much affected. Indeed, we had matched the dumping sites to the closest grid cell where the distance was not more than 0.2°. We also realize that we had mistakenly compared the total amount of "missed" dumping over the 6 considered years to the total annual average of dumping, resulting in the rather large portion of more than 20% "missing" dumping, but actually it is much less. This has been corrected, and all information has been added to the Supplementary Information along with the details on dumping implementation.

Reviewers #3+4 comments	Author responses
The objectives of the study are certainly relevant as the increased of coastal activities, notably dredging and dumping, affects the seafloor integrity, with modification of the sediment structure, water transparency and the cycling of essential elements like carbon and nitrogen with potential effect on carbon sequestration and the emission of greenhouse gasses. The authors have gathered data from various sources on dredging and dumping sites along the continental coast of the North Sea over the past three decades and incorporated them into their modelling setup, which combines several numerical models. This study complements the current understanding of the impact of offshore human activities on sedimentary OC and provides a quantitative regional-scale assessment of dredging and dumping impacts on sedimentary OC. The manuscript is quite well described but could benefit from English editing.	We thank both reviewers for recognizing the relevance of the study. We have addressed all comments in our revised manuscript and provide point-to-point responses below. Responses indicating changes to the manuscript are indicated in italics.
However, the manuscript does not present a significant advance in our holistic understanding of the effect of dredging and dumping activities on the coastal zone. The authors conclude that “ While this study gives first estimates of the magnitudes, further studies gauging impacts of sediment disturbing activities on the overall carbon cycle should include resolving indirect effects”. I fully agree with them. This study is important but is a starting point in the understanding of a complex process. For this reason and for others that I am detailing here below, the level of maturity of the methodology and the scope of the conclusions are not mature enough to influence the thinking in this field. Then, I can not recommend publication of this manuscript in Nature communication and recommend that the authors submit their work in a specialist journal.	Thank you for this comment. To further highlight the novelty of our study and its contribution to a quantitative understanding of the effect of dredging and dumping activities we have reworked the manuscript substantially, performed additional model simulations to better gauge the impacts of material extraction, and added quantifications of natural fluxes for better comparison. We hope to convince the reviewer that the study will be a significant advancement in this field.
1) The mineral sediment is not part of this modelling framework although the authors correctly mention that the aggregate extraction includes a lot of sand and coarse sediment. The capacity of the model to answer the two above-mentioned questions depend on its ability to represent the deposition, erosion and resuspension process of Suspended Particulate Matter (SPM). This capability is not enough assessed in the papers to which the authors refer to. I would have liked to see a clear demonstration of the ability of the model to simulate the SPM, SPM plume and fluxes and, for instance, comparison with satellite data.	The reviewer is incorrect that mineral sediment is not part of the modelling framework, since the initial models used for estimating dredging impacts (Porz et al., 2024; Zhang et al., 2019) do account for mineral (lithogenic) sediment classes. Our newly added material extraction simulations also explicitly account for removal of sand and resuspension of silt and clay. The sediment dynamics in Chen et al. (2025) have been validated against observational data. Note that satellite data is, in general, not recommended for validating SPM fluxes, as most of the SPM flux occurs near the seabed and surface plumes are typically not significant for the overall sediment flux (e.g. Diaz et al.,

	2024; Walsh and Nittrouer, 2009). Therefore, Chen et al. (2025) have used near-bottom SPM measurements for model validation. We do agree with the reviewer that this validation may not hold for other areas, which is the reason why we have done two simulations for dumping with a large range of sinking speeds, which is the most important parameter affecting SPM transport. We therefore believe that the adequate performance of the model in capturing SPM fluxes has been demonstrated and that the remaining uncertainties are well reflected in the study results. We have added more detailed descriptions of the sediment model setup to the Supplementary Information.
2) The impact of aggregate extraction is estimated as the flux of organic carbon out of the sediment at the dredging site. Then the fate of this extracted organic carbon is not investigated. It is possible that this OC is deposited at another place or that it will be degraded in the water column and then outfluxes to the atmosphere. The authors have a model able to estimate that.	We agree and have extended our study by including simulations of aggregate extraction.
3) The impact of dumping is assessed by estimating the place of deposition of OC (the mineral fraction is ignored as for dredging). The authors conclude that the deposition is important in the Estuary where small scale processes are important; The quality of the model in this region is not assessed. The authors refer to a paper by Kossack et al 2023 published in Frontiers in marine research, biogeochemistry. However, having read this Frontiers paper I see some differences mainly the frontiers paper focuses on larger scale. The authors correctly pointed out that the dynamics of sediment is really important in estuaries but the validation of the models in Estuaries where very small scales processes are important is not really addressed in the Frontiers papers,	The mineral fraction of dredged material is now considered in our material extraction model. Note that while Kossack et al. (2023) is indeed aimed at larger scales, that model has been refined for the German Bight in Chen et al. (2025) to resolve smaller scale processes in the estuarine mouth, the Wadden Sea and coastal areas. The aim of the dumping model is not to accurately portray sediment dynamics within estuaries, but to get a general sense of how the dumped sediment will distribute. We have amended the methods section and Supplementary Information to clarify the different model setups.
4) There are a lot of uncertainties in model parameters like for instance the proportion of organic carbon, the liability of organic matter. These uncertainties are correctly pointed out by the authors but their impact on model estimations is not addressed. Considering the limitations of the validation exercise, a kind of ensemble simulations with different liability, OC fraction, sinking speed, mineral versus organic fraction, would have addressed these limitations. Same	Both sediment and OC disturbance are based on measurement data that have high confidence (better than “order of magnitude”), and otherwise we have used upper and lower estimates, as for dumped OC contents and sinking speeds that are key parameters affecting the sediment and POC fluxes. Note that we have used orders of magnitude in Table

with the uncertainty in the distribution of extraction sites mentioned in section 4.1	1 to represent this uncertainty. We have used an average OC content for dumped material based on measurements, but since a conservative tracer was used in the model, a different OC content will merely result in a different scaling of the result, but the relative distribution will remain the same. We have added Supplementary Figure 2 to cover the entire range of feasible OC contents in dumped material. Note that the 3D simulations are computationally very expensive, so performing ensembles with all parameters is not feasible. We have done this sensitivity analysis only for sinking velocity, which is the main parameter governing particle fluxes. Uncertainties in OC content and lability are considered in the total impact numbers (Fig. 3b and Table 1). We are therefore confident in the range of magnitudes given for each of the values. We are also confident that the models produce estimates in the right magnitude, as the models have been validated against field observations in previous studies (Chen et al., 2025; Porz et al., 2024; Zhang et al., 2024; Zhang et al., 2021; Zhang et al., 2019) and are able to reproduce measured sediment OC profiles etc.
5) Chapter 4.3 "Numerical models" is too brief and densely written given the importance of the information it contains, which makes it difficult to follow. The models are scarcely described, with the authors primarily referencing previously published articles, some dating back to the mid-2010s. Have there been any modifications to the model setups since then? A clearer explanation of how the models were used together, supported by a schematic figure and a brief description of the key features of the model configuration, would greatly improve reader comprehension. The implementation of dredging and dumping as modelled processes deserves separate subchapters.	We agree that the models could be better described. We now include more detailed model descriptions in the Supplementary Information for interested readers.
6) A continuation of point 5: If I understand correctly (though it was unclear), the sediment model validation used in this study is presented in Chen, 2025 (ref. 58)? However, the validation in that paper focuses only on suspended particulate matter (SPM) dynamics in a small muddy area, which represents just a fraction of the Wadden Sea. How	We argue that the validity of our dumping simulations does not rely on a correct representation of the inorganic sediment bed dynamics, as our simulations are sinking sediment tracer experiments to get a sense of the fate of dumped material, which will anyhow

can we be confident that the model is capable of accurately simulating dumping in other areas of the North Sea, which are the focus of this study? The sediment bed texture across the North Sea is highly heterogeneous, and so are its mechanical properties—both of which influence bottom fluxes and therefore could significantly affect model results.	have very different properties than the naturally occurring bed. What is important for these simulations is that the hydrodynamics and tracer transport pathways are adequately represented, and these have been demonstrated in Kossack et al. (2023) and Chen et al. (2025). Notably, heterogeneous sediment texture is accounted for in Chen et al. (2025) through different bed roughness based on grain size (now mentioned in Supplementary Methods 2). In addition, the heterogeneous sediment properties of the North Sea are now considered in the newly added model simulations for material extraction. We have added this information to Supplementary Methods 1+2 for clarification.
7) In the abstract, the authors state that they consider dredging and dumping to be important components of the global sediment and carbon budgets. Subchapter 2.3 is dedicated to comparing the impacts of various human activities typical of the North Sea. However, when comparing human-induced impacts with natural sedimentary cycles, the authors rely on estimates from the Wadden Sea as a reference—which represents only a small part of the North Sea. Can these findings be extended to the entire North Sea? And what about at a global scale?	In Table 1 we now compare the amounts to natural fluxes for the entire North Sea. We have added a global perspective to the Discussion (moved from Introduction).
1) Lines 44–50: If the assessment of human activities is not directly compared to the natural marine sediment cycle (e.g., in absolute values or percentages), the significance of their impact is unclear. Riverine input is only one part of natural sediment dynamics; coastal erosion, atmospheric deposition, and other sources should also be considered.	We have added additional estimates for natural fluxes based on literature.
2) Lines 49–50: “It is thus evident that any holistic analysis of contemporary matter fluxes in the Earth system must account for direct human impacts on sediment disturbance.” — This statement may be premature in the Introduction, as it appears to be a conclusion derived later in the manuscript. Consider moving this sentence to the Discussion or Conclusion section where supporting evidence has already been presented.	Agreed, it has been moved to conclusion.
3) Line 60: The recent report identifies dredging and dumping as primary risks to the Wadden Sea’s World Heritage status. What exactly are these risks? How does offshore dumping of additional organic carbon impact benthic communities? It is suggested that these issues be	We have added some specific risks as well as a short discussion on impacts on benthic habitats.

discussed further in the Discussion section, especially since the Heritage status is only briefly mentioned elsewhere in the manuscript without clarification of what aspects are under threat or in need of protection.	
4) Line 69: What role do benthic invertebrates play in this study? Their relevance is not immediately clear and could be explained more explicitly.	We do not discuss the role of benthic invertebrates further in the study. This was meant to convey that dredging essentially pulverizes macrobenthos and ejects their remains into the sea as part of the resulting sediment plume. We have removed that part of the sentence to avoid confusion.
Figure 1 : The caption states: “Colour scheme shows modelled remineralization rate” — Is this output from the current study, or is it based on a previously published dataset or model? This should be clarified. Consider adding a zoomed-in panel showing the Wadden Sea and adjacent Southern Bight, if the figure is intended to highlight human activities in those regions. This would enhance readability and relevance.	This is from Porz et al. (2024). Reference added. Since the Wadden Sea is not resolved in that vulnerability map, we prefer to show the entire North Sea map here.
Table S2: Are the same parameter values (especially critical shear stress and settling velocity) applied to all three sediment classes defined in the model (sand, mixed, mud)? If not, please clarify how these parameters vary by class. If they are the same, this should be explicitly stated and briefly justified.	We have added Supplementary Table 1 listing the parameters for the material extraction experiments.
• Lines 88–96: These paragraphs could be merged into one, as they cover related content and would benefit from improved flow.	Agreed, done.
• Line 97: The value 0.10% is given without a reference. Please provide a citation or clarify how this value was derived.	Reference to Bockelmann et al. (2018) added.
• Lines 99–100: The 87-10-3% partitioning of carbon types (refractory, semi-labile and labile) is introduced without explanation. Please elaborate on the rationale behind this specific partitioning—was it based on literature, model assumptions, or field measurements?	Reference to Porz et al. (2024) added.
• Lines 99–100 & 104–105: There appears to be a contradiction between the statements: "Refractory carbon is unlikely to be remineralized following disturbance." "Refractory carbon is not fully inert ... and may be made more bioavailable when exposed to oxygen and mixed with labile OC." These two points seem to conflict. Consider reconciling them by clarifying under what conditions some remineralization might occur, and whether it is considered significant in the context of your study.	Thank you for pointing this out. We have now expanded this point and added a quantification of this effect.

• Line 107: The phrase “refractory OC remineralization should not exceed a few hundred kt CO₂e/yr” suggests a value that is an order of magnitude higher than your short-term remineralization estimates, yet it’s presented with less emphasis. Please clarify how this value was calculated. What degradation rates in the water column were assumed for disturbed/resuspended refractory OC? It is also unclear whether these estimates are included in Table 1. Please indicate where they are documented or consider adding them.	The numbers have been updated based on the new simulations for aggregate extraction. Degradation rates are now shown in Supplementary Table 1.
Side question: After aggregate extraction, does the exposure of deeper, previously buried sediments to overlying ocean water accelerate OC degradation, for example through enhanced diffusive fluxes (e.g., oxygen penetration)? If so, it might be worth discussing briefly.	Yes, these impacts are considered in the newly added extraction simulations, and this is now mentioned in the Methods.
Line 114: A reference is missing here.	Rephrased to clarify that this is the result of our own analysis.
Lines 123–124: The statement regarding particle retention in the English Channel lacks supporting evidence. The Wilson et al. (2018) sediment maps show that the Dover Strait contains some of the coarsest sediments in the region (gravel fractions 10–70%), suggesting high energy and resuspension. This makes the claim that particles “remain in place after one year” difficult to accept without further clarification. If the area of interest is actually the mud-rich coastal zones of Belgium and the Netherlands, then this should be referred to as the Southern Bight, not the English Channel. Please clarify which area is meant.	Agreed, the phrasing “remains in place” was somewhat misleading since the model does confirm that the Channel hardly retains any dumped material even for the fast-sinking tracer, it just hasn’t travelled quite as far north-eastward as the slow-sinking one. This statement has been rephrased for clarification.
Figure 4: These maps are very interesting and merit more detailed discussion. Consider addressing the following: What is the isobath that marks the edge of observable dumping impact on OC deposition?	The concentration reduces gradually with distance from the dumping sites, so no such isobath can be identified, therefore we prefer not to draw an isobath to avoid misinterpretation.
Why is there a consistent white (zero-deposition) anomaly near the island of Texel across both maps?	This can be explained by the absence of nearby dumping sites as well as elevated bottom shear stresses that do not allow much deposition here. We’ve added Supplementary Figure 3 showing bottom shear stress averaged for one day to illustrate this.
How do the residual currents and tidal mixing unique to this area affect the final distribution of deposited OC?	Added a sentence on the effect of mixing (lack of stratification). Residual north-eastward currents were mentioned already.
If the settling velocity is assumed to be 1 mm/s, how does OC deposition occur in the offshore area between the UK and the Netherlands, which lies tens of kilometers from the nearest dumping site?	This is due to recurrent resuspension and redeposition by the tidal current. Clarified in the text.

What is the baseline distribution of natural mud deposition in the southeastern North Sea? How does the spatial footprint of dumping-related OC deposition compare to natural OC deposition? Which areas are most affected by dumping, and why? Please consider a more explicit spatial comparison in the text or with an additional figure.	We are not aware of any comprehensive field data showing natural OC deposition in the southern North Sea area, except for some point measurements in the mud depocenter near Helgoland (e.g. as mentioned in Chen et al., 2025) and those measured sedimentation rate values represent averages over many decades (e.g. from Pb210 dating) and longer (e.g. from C14 dating). Bioturbation in surface sediments further complicates the interpretation of such “baseline” deposition rates. Quantitatively comparing OC contents (in wt%) from sediment maps to simulated OC deposition rate (in g/m²/yr) would also not be meaningful. Additionally, since the North Sea seabed has been considerably influenced by human activities for over a century, what we can observe from the seabed sediments already contains the human impact. Therefore, to what degree the dumped OC would be integrated into the existing sediment bed cannot be disentangled from field data and can only be roughly estimated by numerical modeling. We have added this point to the discussion.
Chapter 2.3 & Table 1: Line 206: "We show that dredging and dumping can impact the morphodynamics of coastal zones at the same magnitude as natural processes of erosion/sedimentation." — This is a strong claim that is not fully demonstrated in the results. It is only briefly mentioned (see comments on Figure 4). Consider supporting this with a dedicated figure or map showing annual OC deposition from human activities (e.g., dumping) as a percentage of natural OC deposition, with dumping sites overlaid.	Rephrased. This statement refers to the lateral fluxes of sediment and OC in the area based on comparison to estimates for natural processes, which are now added in Table 1.
General: Are there any recommendations for regional dumping site management based on your findings? For example, the result that 22% of dumped mud eventually returns to the same river channels it was dredged from suggests a potentially large inefficiency. This is a striking and policy-relevant insight but is not currently mentioned in the Discussion. Highlighting this could add practical impact to your study.	Note that although not well documented in the scientific literature, this phenomenon is well known in harbour management, at least for the Elbe estuary (“Kreislaufbaggerei”; “circular dredging”) and it is not so much that harbour management is unaware of this issue, but that that it can be much more expensive and/or illegal to dump material farther offshore where it is less likely to return; Only a limited amount of dredged material is dumped offshore. So, we took this more as a confirmation of our model results that this tidal pumping phenomenon is reproduced in our simulations, and we’ve now

	clarified this in the text.
Define blue carbon ecosystems	Rephrased
“the global net sediment flux is due to one or several of these activities” what is meant here? What is the global net sediment flux?	Clarified; we mean the lateral sediment transport
Line 102: please define kt CO ₂ e/yr	Changed to ktCO₂/yr throughout.
Table 1 : Please explain why OWF does not disturb the sediment	It does disturb the sediment, but the cited references contain no estimates for this amount. Clarified that n.d.=no data available
Nutrients in resuspension impact on biology same with the river. Please clarify.	Feedback of nutrient resuspension on biology is not considered here as this is beyond the scope of our study, but it is now mentioned in the Discussion.
Line 97: The average concentration of sediment OC at the extraction sites is 0.10% and from the estimation of sand and gravel estimate the amount of organic carbon extracted. First, this 0.1 % seems to be highly empirical and is poorly validated	This number is based on overlaying the extraction areas with the published sediment map of Bockelmann et al. (2018). We’ve now added a reference to that.
Lines 103-105: For comparison, the recent fuel emissions of the European dredging fleet was estimated at 600-800 kt CO ₂ e/yr according to an industry report. Nevertheless, it is important to note that the refractory OC fraction is not fully inert, and may be made more bioavailable when exposed to oxygen, especially when mixed with more labile compounds. This is not clear how the authors have taken this uncertainty into account.	We’ve added a reference to Sanches et al. (2021), who estimated that effect to a 21-65% increase in remineralization rates. The results have been updated according to our new model simulations.
Line 116-117: carbon contained in dumped material is in the range of 500-5,000 kt/yr, reflecting the uncertainty in the OC content of dumped material. How does this uncertainty be taken into account?	The uncertainty has been taken into account in our results, as evidenced by the range of magnitudes given for carbon disturbance by dumping in Table 1, as well as the grey shading in Fig 3b.

Reviewer #5 comments	Author responses
The manuscript “Dredging and dumping impact coastal fluxes of sediment and organic carbon” by Porz and coauthors builds on earlier work by the team to assess turnover of carbon on the continental shelf of the North Sea by anthropogenic activities. In Zhang et al 2024 (https://doi.org/10.1038/s41561-024-01581-4), the authors evaluated the impact of bottom trawling on carbon remineralization and release to the atmosphere. Using a similar approach and modeling environment, in this study the authors evaluate the carbon consequences from aggregate extraction and material dumping. This is an interesting modeling study and below I highlight questions and places to strengthen the manuscript:	We thank the reviewer for their useful comments which have helped us improve the manuscript. We address all comments below. Responses indicating changes to the manuscript are indicated in italics.
Abstract: It is a bit disingenuous to start the abstract noting the lack of large scale estimates of the impact of anthropogenic activities on coastal sediment carbon dynamics given the spate of recent articles published, including by this author team. [e.g. https://doi.org/10.1038/s41586-023-06014-7, https://doi.org/10.5194/bg-21-2547-2024, 2024, https://doi.org/10.3389/fmars.2023.1125137, https://doi.org/10.1016/j.wasman.2021.11.031, and others]	Agreed. We have rephrased the abstract and introduction to acknowledge the previous work on bottom trawling. Though the work by Svensson et al. on dumping is very interesting, it only indirectly relates to sediment and carbon disturbance, and considering the number of references allowed by the journal is limited, we decided not to include it.
Introduction: Figure 1 is introduced here with mineralization rates shown. Are these rates attributed to natural processes plus anthropogenic disruption due to extraction and dumping or are these background rates from the model without those impacts? If these rates are part of the findings of the paper, it would be odd to only mention Figure 1 in the Introduction.	These are the background rates produced by the model and a map showing that vulnerability data has been published in Porz et al. (2024), so we do not present it as a new result, but have now added reference to that study.
Results: Better referencing and context is needed to support the amount of carbon available to release to the atmosphere. Comparison with in situ measured rates of benthic oxygen demand/utilization would be useful as well.	We have rephrased to clarify that these are aqueous emissions and added a part on buffering capacity and potential portion that is equilibrated with the atmosphere to the discussion.
“The average concentration of sediment OC at the extraction sites is 0.10%.” Reference?	Reference to Bockelmann et al. (2018) added.
“In our model, 87% of this disturbed OC constitutes the refractory fraction, which is unlikely to be remineralized following disturbance, while 10.0% and 3.0% of OC are semi-labile and labile...” Reference?	Reference to Porz et al. (2024) added.
“Nevertheless, it is important to note that the refractory OC fraction is not fully inert, and may be made more bioavailable when exposed to oxygen,	Typical extraction depth is 30 cm, as mentioned in the Methods. Oxygen penetration is dependent on sediment

especially when mixed with more labile compounds³⁴” What is the depth of oxygen penetration considered in the remineralization model? What is the depth of sediment typically extracted? Typically sandy sediments have advective transport and mixing down to ~15 cm, resulting in regular exposure to oxygen. [https://doi.org/10.1016/j.earscirev.2022.103987]	composition and grain size and is deeper for more coarser sediments under the same hydrodynamic conditions, below which degradation rates are reduced exponentially with depth. The O₂ penetration depth in the sandy seabed is normally a few cm according to field observation in the North Sea (Zhang et al., 2021). This has been added to the more detailed model description in Supplementary Methods 1.
Results: Given the potential variability in some of these model parameters, it would be useful to have a deeper discussion about sources of uncertainty and present more broadly a sensitivity analysis and potential impact on values in Table 1. It is good those values are order of magnitude, but within the context of the manuscript it would be good to understand how certain they are and how sensitive they are to some of the model assumptions.	We are confident in the range of magnitudes given for each of the values. Both sediment and OC disturbance are based on measurement data that have fairly high confidence (better than “order of magnitude”), and otherwise we have used upper and lower estimates, as for dumped OC contents and sinking speeds that are most critical for estimating transport fluxes. Note that the 3D simulations are computationally very expensive, so performing ensembles with all parameters is not feasible and we performed sensitivity analysis mainly on the two previously mentioned key parameters in this study. Nevertheless, we are also confident that the models produce estimates in the right magnitude, as the models have been validated against field observations in our previous studies (Zhang et al., 2019, 2021, 2024; Porz et al., 2024; Chen et al., 2025) and are able to reproduce measured sediment OC profiles etc. For the dumping simulations, we now include min and max ranges for the OC contents in dumped material in Supplementary Figure 2.
Discussion: “For example, metabolic processes may generate 16 225 not only dissolved inorganic carbon, but also alkalinity, which under some circumstances can strengthen the coasts role as a carbon sinks⁴⁷.” The impact of changes in organic matter remineralization on organic and inorganic carbon and alkalinity fluxes is definitely important. Alkalinity is typically generated during anoxic remineralization reactions (see https://linkinghub.elsevier.com/retrieve/pii/B9780323997621000322 for a good review). What mechanisms do the authors think would lead to increased anoxia in bottom sediments with extraction and dumping compared to no disturbance conditions? A recently published paper indicated that enhanced oxygenation may lead to pyrite oxidation, resulting in loss of alkalinity and acidification enhancement	Agreed. This was in reference to ports being sources of alkalinity, but we agree that this is not specific to sediment disturbance, so we removed the corresponding reference and added two on pyrite oxidation.

[<https://www.science.org/doi/10.1126/sciadv.adp9112>], counter to the production invoked in this manuscript.

References

- Bockelmann, F.-D., Puls, W., Kleeberg, U., Müller, D., Emeis, K.-C., 2018. Mapping mud content and median grain-size of North Sea sediments – A geostatistical approach. *Mar. Geol.* 397, 60–71. <https://doi.org/10.1016/j.margeo.2017.11.003>.
- Chen, J., Zhang, W., Porz, L., Arlinghaus, P., Hanz, U., Holtappels, M., Schrum, C., 2025. Physical Mechanisms of Sediment Trapping and Deposition on Spatially Confined Mud Depocenters in High-Energy Shelf Seas. *J. Geophys. Res. Oceans* 130, e2025JC022622. <https://doi.org/10.1029/2025JC022622>.
- Diaz, M., Grasso, F., Sottolichio, A., Le Hir, P., Caillaud, M., 2024. Investigating sediment dynamics on a continental shelf mud patch under the influence of a macrotidal estuary: A numerical modeling analysis. *Across the sediment-water interface: Biogeochemical cycling in coastal and shelf seas* 282, 105334. <https://doi.org/10.1016/j.csr.2024.105334>.
- Porz, L., Zhang, W., Christiansen, N., Kossack, J., Daewel, U., Schrum, C., 2024. Quantification and mitigation of bottom-trawling impacts on sedimentary organic carbon stocks in the North Sea. *Biogeosciences* 21, 2547–2570. <https://doi.org/10.5194/bg-21-2547-2024>.
- Sanches, L.F., Guenet, B., Marino, N.d.A.C., de Assis Esteves, F., 2021. Exploring the Drivers Controlling the Priming Effect and Its Magnitude in Aquatic Systems. *J. Geophys. Res. Biogeosci.* 126, e2020JG006201. <https://doi.org/10.1029/2020JG006201>.
- Siham, K., Fabrice, B., Edine, A.N., Patrick, D., 2008. Marine dredged sediments as new materials resource for road construction. *Waste Management* 28, 919–928. <https://doi.org/10.1016/j.wasman.2007.03.027>.
- Verlaan, P., 2000. Marine vs Fluvial Bottom Mud in the Scheldt Estuary. *Estuar. Coast. Shelf S.* 50, 627–638. <https://doi.org/10.1006/ecss.1999.0599>.
- Walsh, J.P., Nittrouer, C.A., 2009. Understanding fine-grained river-sediment dispersal on continental margins. *Mar. Geol.* 263, 34–45. <https://doi.org/10.1016/j.margeo.2009.03.016>.
- Zhang, W., Neumann, A., Daewel, U., Wirtz, K., van Beusekom, J.E.E., Eisele, A., Ma, M., Schrum, C., 2021. Quantifying Importance of Macrobenthos for Benthic-Pelagic Coupling in a Temperate Coastal Shelf Sea. *J. Geophys. Res. Oceans* 126, e2020JC016995. <https://doi.org/10.1029/2020JC016995>.
- Zhang, W., Porz, L., Yilmaz, R., Wallmann, K., Spiegel, T., Neumann, A., Holtappels, M., Kasten, S., Kuhlmann, J., Ziebarth, N., Taylor, B., Ho-Hagemann, H.T.M., Bockelmann, F.-D., Daewel, U., Bernhardt, L., Schrum, C., 2024. Long-term carbon storage in shelf

sea sediments reduced by intensive bottom trawling. *Nature Geoscience*.

<https://doi.org/10.1038/s41561-024-01581-4>.

Zhang, W., Wirtz, K., Daewel, U., Wrede, A., Kröncke, I., Kuhn, G., Neumann, A., Meyer, J., Ma, M., Schrum, C., 2019. The Budget of Macrobenthic Reworked Organic Carbon: A Modeling Case Study of the North Sea. *J. Geophys. Res. Biogeosci.* 124, 1446–1471. <https://doi.org/10.1029/2019JG005109>.

Reviewer #1 comments	Author responses
I have read the new version of the manuscript, and I am happy with how the authors have addressed all comments. I have a few small suggestions for improving clarity, but I think this manuscript is a great addition to the scientific literature and will attract a lot of attention from the scientific and policy-making communities.	We thank the reviewer for their insightful comments that have helped us greatly in improving the quality of our manuscript. We have addressed all comments in our revised manuscript and provide point-to-point responses below. Responses indicating changes to the manuscript are indicated in italics.
L78: define 'aqueous CO2 emissions' for non-specialists	Defined: "... , or about 3.67 ktCO₂ released to the seawater (aqueous emissions) if all OC is remineralised to CO₂,..."
L84: 21-65% relates to the total mineralisation rate or the mineralisation of the unreactive fraction alone? If it is the unreactive fraction, you would have to first get a rate for the unreactive fraction, and then calculate the increase – not just multiply the whole rate by 1.65. In any case, it is a bit unclear if phrased this way.	Since the priming effect is usually operationally defined as an increase in remineralisation rate of the bulk material (i.e., the mixture), the relative contributions of different reactivity fractions have not been separated in the cited studies. According to Sanches et al. (2021), this increase is 23.4%–91.6% (21-65 was the range of the log-response ratio). Our upper limit of 10 ktCO₂/yr for mining still holds under this assumption. We have corrected and re-worded this to clarify that it is the bulk sediment OC that is affected.
L224: I know I mentioned 0.6 in relation to the buffering capacity of seawater – but that was a mistake on my part – 0.6-0.7 related to the CO₂ release for calcium carbonate precipitation. In perfect equilibrium with the atmosphere, and assuming the pCO₂ in the atmosphere does not change, a DIC added to the water-column will always degass, since the alkalinity and pCO₂ set the DIC concentration. The 0.6-0.7 relates to the fact that not all water will re-equilibrate with the atmosphere due to circulation, and is thus a global number – which is not the same as the buffering capacity.	Thank you for the clarification. Rephrased: "It is also important to note that the numbers for excess carbon remineralisation presented here should not be equated to atmospheric CO₂ emissions, as only a portion of remineralised OC will directly outgas due to the buffering capacity of the seawater carbonate system. Nevertheless, a corresponding effect on air-sea CO₂ exchange is likely, as eventual equilibration with the atmosphere is expected in the shallow, mixed coastal zone."
L238: I am not sure what riparian means, and I suspect I am not alone – maybe replace with a more common word?	Changed to "...other nations bordering the North Sea."

Reviewer #3 comments	Author responses
The revised article has improved in quality, but I argue that with this advanced model and the extensive and detailed databases for dredging and dumping, the manuscript needs to expand the analysis of simulation results and balance it between both the Wadden Sea and the Southern Bight as two distinct regions. Model runs with two different sinking speeds of particles are not enough for me to believe that this process is as important for carbon disturbance as rivers, even on the scale of the Wadden Sea. I understand the challenges coming from quantification and parameterization of processes as dredging and dumping, but with a model like that their impacts should be analyzed deeper for the purpose of transferability of model results:	We thank the reviewer for their insightful comments that have helped us greatly in improving the quality of our manuscript. We have addressed all comments in our revised manuscript and provide point-to-point responses below. Responses indicating changes to the manuscript are indicated in italics.
1. You model the entire North Sea, but the results disproportionately focus on the Wadden Sea, while the Southern Bight, which is equally busy in terms of dumping/dredging, is almost ignored or counted as a part of the greater North Sea (e.g. “Wadden Sea” or “German Bight” are mentioned 24 times in the manuscript, “Southern Bight” and “Southern North Sea” (which I believe also includes the Wadden) - less than 10). What are the differences between the Wadden Sea and SBNS for the fate of dumped carbon? How hydrodynamics, composition of the sediment bed and local biogeochemistry shape the fate of the dumped matter? Can Wadden sea depocenter absorb some scattered deposited material?	We understand the reviewer’s concern that we focus disproportionately on the Wadden Sea. We now distinguish both areas more clearly when describing the dumping results (l.121 ff.) and added a general distinction of non-depositional vs. depositional areas to the discussion (l.250 ff.). Nevertheless, we would like to keep the focus of the analysis on the Wadden Sea for three reasons:  1. In contrast to the Southern Bight, the Wadden Sea is a net sediment sink, allowing a quantitative comparison of sediment and OC deposition between natural and human impacts. 2. As a UNESCO World Heritage Site, the Wadden Sea is a protected area threatened by human impacts. 3. The Wadden Sea has a higher spatial resolution in our model, allowing a better distinction of the fate of dumped material, e.g. within tidal flats and basins.

2. The flocculation process, that is disproportionately important for dumping, is implemented in the model with a sinking speed x20 than a real one. What is the source of this value? Regardless, I would either create another class of flocculated mud and define flocculation intensity as a function of mud content in the water, or, alternatively, directly parameterize the sinking speed as this function. It will allow a more realistic spread.	Maximum sinking velocities for SPM in the study area have been measured at about 0.7 mm/s (Maerz et al., 2016), close to our maximum value of 1 mm/s. We have added this information to the manuscript. The sediment module currently does not allow for variable sinking speeds. In order to account for the exceptionally fast sinking of aggregates of dumped material, we now performed an additional simulation wherein the dumped material was injected at the bottom water layer rather than the surface layer, as monitoring of dumping processes have found that the dumped material initially collapses quickly at the seafloor and the amount remaining in suspension directly after initial disposal is only a few percent (Gundlach et al., 2024). The results have been added to the manuscript (see Table 1, Fig. 4). The resulting distribution is in between the rapidly and slowly sinking tracers when injected at the surface, but closer to that of using rapidly sinking tracers, especially when comparing the amounts deposited near the coast (Table 1). We believe this provides a good sense of the range of expected distributions.
3. In Supplement: “For the dumping experiments, the seafloor is left bare, i.e., no sediment is initialized in the model”. Does your model account for different properties of sand and mud in terms of deposited carbon retention (or it’s only the shear stress)? Does your model (in this dumping experiment) simulate the natural carbon cycle (it was not clear for me)? Bioturbation/bioirrigation brings carbon into deeper layers, but if naturally deposited carbon is absent, irrigation will exclusively act on the dumped material, overestimating its degradation and burial.	As the dumping experiments use inert sinking tracers (as stated in l. 420), the incorporation of dumped material into existing seabed is not considered. We hope our newly added conceptual sketch (Fig. 6) makes this clear.
4. It would be really interesting to see a relative impact (%) of dumping compared to the natural deposition as it is in the North Sea, with a significance horizon of	As stated in our earlier reply, there is no spatial data on natural deposition in the Southern North Sea, as most of the Southern North Sea is non-depositional;

5%. It will show how localized the impact of dumping is, regardless of its absolute contribution to the carbon budgets on the regional scale. Same stands for dredging.	we therefore chose to focus on the Wadden Sea, which is a net sediment sink where the relative impact of dumping can be assessed. In the Wadden Sea, we compare our dumping results quantitatively to the total mud budget according to existing estimates. We have added this explanation to the results for clarification. For dredging, we have added a Supplementary Fig. 1b showing the relative change to sediment OC due to extraction, since we do have OC maps covering all extraction areas. Note, however, that this is not the same as a change in deposition rate, since we only compare to the existing sediment bed, but we think it gives a sense of the relative impact.
5. I thought about sensitivity tests for different proportions of sediments and carbon in the dumped material, but you have created a long realistic database. In this case, how are the sites where OC-rich mud is dumped, different from OC-poor sandy sediment sites? I believe that if we go into details of dumping materials, an analysis or elaboration should be done about impacts of those different materials.	We have now added Supplementary Fig. 4 showing the estimated proportions of fine and coarse material. The proportions of different materials is considered in our data analysis by distinguishing between coarse and fine-grained material. However, there is little data on the proportion of OC in the dumped material, which is why we have used upper and lower limits, as reflected in the time series in Fig. 3b as well as the sensitivity tests in Supplementary Fig. 2. We only consider dumping of fine-grained material (including OC) in our model, since the model has been validated for suspended particulate matter transport, which has a spatial effect far beyond the dumping sites, while coarse-grained material (sand/gravel) is mainly transported as bedload with only a local effect. We have added this limitation to the discussion (l. 310 ff.).
6. Methodology: it needs a comprehensive sketch of the processes, represented in the model, with some additional info about sediment classes, organic carbon content range etc. As a	This is a good suggestion. We have added a conceptual figure (Fig. 5) sketching the processes considered in the model, as well as Table 3 with an overview of the different model setups.

reader, I had a hard time switching constantly between explanation in methodology chapter, supplement and numerous references to the previous setups to understand the modelling logic of the setup.	
7. Figure 1 (minor): Zoom-in is still needed for the relevant regions: the SBNS and the Wadden Sea to demonstrate the sites. I didn't understand your comment about the Wadden sea and vulnerability map.	The reason we chose not to zoom in on the Southern North Sea and Wadden Sea in Fig. 1 is because there are no models or data for the OC vulnerability in the Wadden Sea (note that the values for bulk remineralisation in the Wadden Sea are left blank in the map in Fig. 1; we now clarify this in the description). Instead, we show the dumping activity in a zoom-in with more detail (without bulk remineralisation) in Fig. 3a.
The current analysis for the North Sea is quite region-specific thus limiting manuscript's relevance to mostly the North Sea. Expanding the analysis will help a reader from any part of the world to project the findings on their region of interest and get first-hand estimates without a need to run a coupled model. Therefore I believe, addressing this criticism will align the manuscript with the standards of Nature.	We thank the reviewer for this suggestion. We agree that a global perspective would enhance the manuscript. We have added global estimates by extrapolating our results to available global data for dredging and dumping and added this to the discussion section and abstract, as well as an additional figure (Fig. 5) with a global distribution of dredging.

References

- Gundlach, J., Herbst, M., Alex, A.S., Zorndt, A., Jordan, C., Visscher, J., Schlurmann, T., 2024. Simulating the near-field dynamic plume behavior of disposed fine sediments. *Front. Mar. Sci.* 11. <https://doi.org/10.3389/fmars.2024.1416521>.
- Maerz, J., Hofmeister, R., van der Lee, E.M., Gräwe, U., Riethmüller, R., Wirtz, K.W., 2016. Maximum sinking velocities of suspended particulate matter in a coastal transition zone. *Biogeosciences* 13, 4863–4876. <https://doi.org/10.5194/bg-13-4863-2016>.

Reviewer #3 comments	Author responses
I think the manuscript is well-written and all my notes were taking into account. I just remind to be aware, that the SBNS is not as non-depositional as it might appear in some setups. Of course it's not the Trench, but De Borger's campaign have shown OC content in the sediment bed. Some of the PP is deposited during the leap tide and is not resuspended back. I also highlight very good schematics and images. I think the authors have done a great work on understanding the processes of dredging and dumping and I recommend the manuscript for the publication. Please consider the notes down below as suggestions to improve the manuscript.	We thank the reviewer for their insightful comments that have helped us greatly in improving the quality of our manuscript. We have addressed all remaining comments in our revised manuscript and provide point-to-point responses below. Responses indicating changes to the manuscript are indicated in italics.
Add explicit statement that dumping material is inert, no OC remineralisation included.	Added to the methods section
Add justification why dumping uses only one tracer class while extraction uses six.	Added to the methods section.
Insert mechanistic hydrodynamic contrast between Wadden Sea and Southern Bight.	Included.
Add fine = OC-bearing explanation in dumping introduction.	Added.
Add global assumptions link (OC content) to Methods.	As the global upscaling is part of the discussion and quite straight-forward, we believe it is easy enough to follow without a dedicated section in the Methods and prefer to leave it as is.
Clarify circular dredging does not bias spatial patterns.	Explanation added.
Add note on log-scale inflation in Fig. 4 caption.	"Note the logarithmic colour mapping" included.
Add tidal basin definition reference near Table 1.	Added to table description.
Add hydrographic year mismatch explanation (2000 vs 2012).	Added to Table 3 description ("The simulation periods are chosen to coincide with periods for which the respective model setups have been validated.")
Add 0.2° exclusion rule explanation to main Methods.	Moved to main methods.

SUPPLEMENT	
Clarify fine/coarse OC roles in Table 1.	Added to table description of Suppl. Table 1.
Add “bare seafloor” explanation.	Included in Suppl. Methods 2: “For the dumping experiments, the seafloor is left bare, i.e., no sediment is initialized in the model, as we are only interested in the redistribution of dumped material.”
Justify 0.2° cutoff.	Added: “This cutoff is chosen such that upstream and inshore dumping activities are discarded, while including nearshore or estuarine dumping activities that lay just outside the model domain.”
Clarify $C = 296 \text{ kg/m}^3$ derivation.	Derivation added.
Add rationale for 0.05 and 1 mm/s.	Reference to measurements by (Maerz et al., 2016) added.
Add log-scale note for Supplement Fig. 2.	Included.
Add explanation for Dec 30, 2014 shear stress date.	This is the last day of the hydrodynamic simulation; The hydrodynamic fields are not saved continuously to save disk space; note added.
Link Supplement Fig. 4 to OC-bearing logic.	Added.
FIGURES	
Add missing Wadden Sea blank-vulnerability note to Fig. 1 caption.	Note added.
Add midpoint OC-range information to Fig. 3a/b captions.	Added.

References

- Maerz, J., Hofmeister, R., van der Lee, E.M., Gräwe, U., Riethmüller, R., Wirtz, K.W., 2016. Maximum sinking velocities of suspended particulate matter in a coastal transition zone. *Biogeosciences* 13, 4863–4876. <https://doi.org/10.5194/bg-13-4863-2016>.